# Atomically engineering interlayer symmetry operations of two-dimensional crystals

Ziyi Han[1,2,9], Shengqiang Wu[1,9], Chun Huang[3,9], Fengyuan Xuan[4], Xiaocang Han[1], Yinfeng Long[5], Qing Zhang[2,6,7], Junxian Li[1], Yuan Meng[1], Lin Wang[5], Jiahuan Zhou[8], Wenping Hu[2,6], Jingsi Qiao[3], Dechao Geng[2,6,7] ✉ & Xiaoxu Zhao[1] ✉

Crystal symmetry, which governs the local atomic coordination and bonding environment, is one of the paramount constituents that intrinsically dictate materials' functionalities. However, engineering crystal symmetry is not straightforward due to the isotropically strong covalent/ionic bonds in crystals. Layered two-dimensional materials offer an ideal platform for crystal engineering because of the ease of interlayer symmetry operations. However, controlling the crystal symmetry remains challenging due to the ease of gliding perpendicular to the Z direction. Herein, we proposed a substrate-guided growth mechanism to atomically fabricate AB'-stacked $SnSe_2$ superlattices, containing alternating $SnSe_2$ slabs with periodic interlayer mirror and gliding symmetry operations, by chemical vapor deposition. Some higher-order phases such as 6 R, 12 R, and 18 C can be accessed, exhibiting modulated nonlinear optical responses suggested by first-principle calculations. Charge transfer from mica substrates stabilizes the high-order $SnSe_2$ phases. Our approach shows a promising strategy for realizing topological phases via stackingtronics.

The perception of symmetry is paramount in dictating the fundamental regularities of nature, while the diverse textures of the world arise from mechanisms of symmetry breaking[1]. Engineering symmetry is significant to understanding the various guises of nature. In particular, at microscopic scales, various applications of matter in fields such as magnetism, superconductivity, many-body physics, electronics, and optics can be understood and engineered by the preservation or breaking of symmetry[2]. For well-studied topologically protected semimetals, fermionic quasiparticles are directly relevant to the degeneracy of band crossing points near the Fermi level, which is determined by the crystalline symmetry[3]. The Dirac semimetal can transform into a Weyl semimetal when symmetry is broken, due to the splitting of Dirac points[4]. Additionally, symmetry-protected topological phases vanish when the protecting symmetry is broken. A new paradigm of crystal symmetry has recently emerged in the study of materials' electronic properties[5]. Janus 2D monolayers with broken mirror symmetry, exhibit asymmetric electron distribution that potentially enhances charge density near the Fermi level. These unique 2D manifolds boost their activities in the oxygen reduction reaction[6] and offer the potential for high-frequency neural stimulation applications[7]. These materials possess low activation barriers and high diffusion coefficients, making them exceptionally suitable for lithium-

[1]School of Materials Science and Engineering, Peking University, Beijing 100871, China. [2]Key Laboratory of Organic Integrated Circuits, Ministry of Education & Tianjin Key Laboratory of Molecular Optoelectronic Sciences, Department of Chemistry, School of Science, Tianjin University, Tianjin 300072, China. [3]MIIT Key Laboratory for Low-Dimensional Quantum Structure and Devices & Advanced Research Institute of Multidisciplinary Science, Beijing Institute of Technology, Beijing 100081, China. [4]Suzhou Laboratory, Suzhou 215123, China. [5]School of Mechanical Engineering, Shanghai JiaoTong University, Shanghai 200240, China. [6]Collaborative Innovation Center of Chemical Science and Engineering (Tianjin), Tianjin 300072, China. [7]Beijing National Laboratory for Molecular Sciences, Beijing 100190, China. [8]Wangxuan Institute of Computer Technology, Peking University, Beijing, China. [9]These authors contributed equally: Ziyi Han, Shengqiang Wu, Chun Huang. ✉e-mail: gengdechao_1987@tju.edu.cn; xiaoxuzhao@pku.edu.cn

ion battery anodes. Moreover, the modulation of crystal symmetry has enabled materials to exhibit unique optical responses, expanding their utility in terahertz and mid-infrared detection[8].

However, current research efforts are predominantly focused on three-dimensional nanoclusters and their assembled heterostructures, with metallic (e.g., Au, Ag, Pt, and Ru) and alloy (e.g., PdAg, PtAg, TaAs, and NbP) nanomaterials considered as exemplary cases[9,10]. The variation in the crystal symmetry is highly correlated to the electronic structures. For example, hcp-Ni crystals ($60 \mu\Omega$ cm) demonstrate significantly higher resistivity compared to fcc-Ni film ($6.9 \mu\Omega$ cm)[11]. The catalytic properties of hcp-Co are superior to $\varepsilon$-Co crystals, primarily owing to the rich electronic density near the Fermi level[11]. So far, there are limited methods to tune crystal symmetries, such as wet-chemical reduction[12], seed-mediated epitaxial growth[13], thermal annealing[14], vapor-liquid-solid[15], etc.[16,17]. However, the precise design of crystal symmetry, particularly at the atomic scale, targeting metastable or nonstable phases remains challenging.

2D van der Waals (vdW) layered materials belong to a class of highly anisotropic materials, featuring strong intralayer covalent bonds but weak interlayer vdW interaction. The weak vdW coupling offers an additional degree of freedom to modify the crystal symmetry via interlayer gliding and mirror symmetry[18]. This field of research, termed stackingtronics[19,20], involves interlayer shifts and orientation angles within layers. Stacking engineering has been demonstrated effective in manipulating crystal symmetry, giving rise to correlated unprecedented properties including nonlinear optics (NLO)[21,22], spin-obit physics[23,24], piezoelectricity[25,26], and superconductivity[27,28]. Specifically, AB-stacked bilayer $h$-BN exhibits intriguing out-of-plane ferroelectric polarization in contrast to the conventional AA'-stacked $h$-BN[29]. Similarly, hexagonal $MoS_2$ bilayers exhibit sliding ferroelectricity through altering the stacking sequence from centrosymmetric AA' into AB stacking polytype[30,31]. Transition metal trihalides, like $CrI_3$, present stacking registry-dependent magnetism. The rhombohedral $R\bar{3}$ phase taking an ordered ABC stacking order along the armchair direction displays ferromagnetic properties. In contrast, the monoclinic C2/m (ABC stacking along the zigzag direction) is characterized by their antiferromagnetic behavior[32,33]. Analogous transition metal phosphorous trichalcogenides ($MPX_3$) are recognized for their electronic and magnetic properties, which vary with stacking configurations (C2/m and $R\bar{3}$ phases), affecting the spin states of transition metal ions and directly determining the exchange energy[34,35]. Moreover, recently discovered layered $MnBi_2Te_4$ has exhibited strong stacking-dependent magnetic properties. Stacking sequences such as AB, AC, AB', and AA' predominantly exhibit an antiferromagnetic ground state, whereas configurations like AA and AC' stackings are predisposed towards a ferromagnetic phase[36]. These phenomena underscore the complexity of interlayer interactions and their impact on material properties.

So far, top-down methods have been widely employed for accurate control over the stacking sequences[37]. However, challenges such as polymer residues, low yield, and interlayer contamination significantly affect the interlayer coupling, thereby hindering applications that require high-quality materials[38]. Conversely, bottom-up methods, e.g., chemical vapor deposition (CVD), offer a superior platform to directly synthesize vdW materials with sharp interfaces, assisted by salt catalysts[39], thermal energy[40,41], and the introduction of extra intrinsic atoms[42]. Specifically, sodium chloride is employed as a promoter to reduce the nucleation energy barrier, in which AA-stacking $MoS_2$ flakes with a ratio of up to 99.5% have been achieved[39]. A two-step CVD growth process was adopted with reverse gas flow to ensure enough kinetic energy triggering the growth of the bilayer, obtaining AB and AA stacking registries[40]. Furthermore, incorporating Mo as interstitial atoms in the hollow position can induce a transition from AA' stacking to the thermodynamically unstable AB' polytype[42]. Certainly, more properties dependent on the stacking polytypes of the

aforementioned materials remain undiscovered. Meanwhile, achieving the atomic-scale synthesis of large-area vdW materials featuring higher-order superlattices and controlled interlayer symmetry operations remains a vital challenge.

Herein, we proposed a substrate-guided growth mechanism to synthesize $SnSe_2$ with a controlled stacking order down to the single-layer symmetry precision. As demonstrated in T-polymorphic-based $SnSe_2$, ordered AA and AB' stacking registries can be regulated by the charge transfer from the mica substrate and subsequently stabilized by local metal-rich chemical potentials. Highly crystalline $SnSe_2$ crystals with distinct stacking polytypes are verified by scanning transmission electron microscopy (STEM) and Raman spectra. The underlying growth mechanism, namely substrate-directed, for synthesizing various configurations has been proposed. The substrate-mediated and controlled chemical potential of the iodide precursor synergistically dictates the $SnSe_2$ slabs, i.e., well-ordered interlayer gliding and/or inverted symmetry, leading to the long-range ordering of $SnSe_2$ superlattices, e.g., 6 R, 12 R, 18 R, and 18 C. Characterized by periodic inversion symmetry and interlayer gliding operation, higher-order phases exhibit symmetry-dependent NLO responses. This phenomenon, coined as stackingtronics, has also been consistently observed in other T-phase transition metal dichalcogenides (TMDs), such as $TiSe_2$, providing an alternative approach to modulate crystal symmetry-dependent topological phases in two-dimensional vdW crystals.

## Results and discussion
### Selective growth of AA and AB'-stacked $SnSe_2$ crystals

A general ambient pressure CVD route was employed to synthesize $SnSe_2$ crystals, as illustrated in Fig. 1a. Selenium (Se) and tin Iodide ($SnI_2$) powders were employed as chalcogen and metal precursors, respectively, with mica as the growth substrate. It is known that the sublimation temperature of metal precursors, determining the metal chemical potential, are key parameters to control the phase of 2D materials[43]. Accordingly, the $SnI_2$ precursor was chosen for the ease of decomposition at elevated temperatures, benefiting the formation of a more stable and uniform growth environment[44]. Meanwhile, we observed the temperature-dependent crystal shape transformation from sharply defined triangles to rounded edges in $SnSe_2$, eventually culminating in fully rounded forms (Supplementary Fig. 1). When the sublimation temperature was set at 320 °C, triangular-shaped single-crystalline $SnSe_2$ flakes were obtained. On the other hand, when the growth temperature was elevated to 340 °C reaching a metal-rich condition, the edges become rounded (Fig. 1b and Supplementary Fig. 2), suggesting a nonequilibrium growth condition between dynamics and thermodynamics[45]. When the substrate temperature was further increased to 750 °C, the obtained crystals predominantly consisted of SnSe (Supplementary Fig. 3). The thickness of the as-grown $SnSe_2$ flakes typically ranges from bilayer (-1.4 nm) to tens of nanometers through modulating the gas flow of $H_2$, as confirmed by atomic force microscopy (AFM) (Supplementary Fig. 4).

The two types of $SnSe_2$ flakes grown at different conditions are highly crystalline, as evidenced by X-ray diffraction (XRD), selected area electron diffraction (SAED) (Supplementary Fig. 5), and Raman spectra. XRD measurement exhibits remarkable (00$l$) reflections from both $SnSe_2$ and mica substrate, indicating excellent out-of-plane alignment (Supplementary Fig. 6). Characteristic Raman peaks at approximately 115.4 cm$^{-1}$ and 185.2 cm$^{-1}$, corresponding to the $E_{2g}$ and $A_{1g}$ vibrational modes[46], respectively, serve as fingerprints to identify $SnSe_2$ and confirm the absence of other by-products, e.g., SnSe (Fig. 1d). It is known that Raman spectroscopy is insensitive to stacking sequences[47]. Therefore, second harmonic generation (SHG) measurements are employed to unveil the crystal symmetry of the as-grown $SnSe_2$ crystals. Typically, conventional T-phase TMDCs, adopting an AA interlayer stacking registry, belong to P$\bar{3}$m1 space group (Fig. 1c). Consequently, AA-stacked T-phase TMDCs are optically dark for NLO

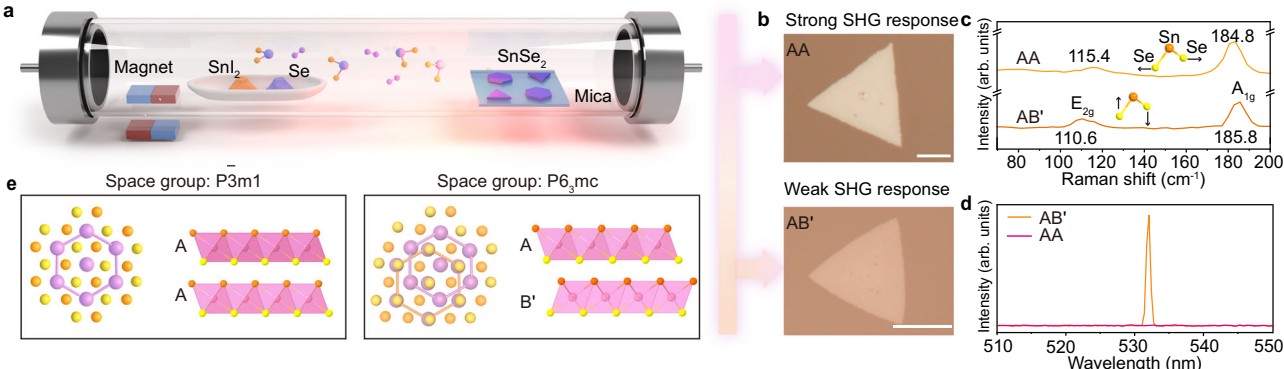

**Fig. 1 | Controlled synthesis of AA- and AB'-stacked SnSe₂.** **a** Schematic illustration of the CVD setup for the phase control in SnSe₂. Optical images (**b**), Raman spectra (**c**), and SHG (**d**) of AA- and AB'- stacked SnSe₂ flakes, respectively. **e** Atomic models (top and side views) of AA- and AB'-stacked SnSe₂. Pink atoms represent Sn atoms, the orange atoms represent the upper-layer Se atoms bonded to Sn, and the yellow spheres represent the lower-layer Se atoms bonded to Sn. Scale bar: **b** 10 μm.

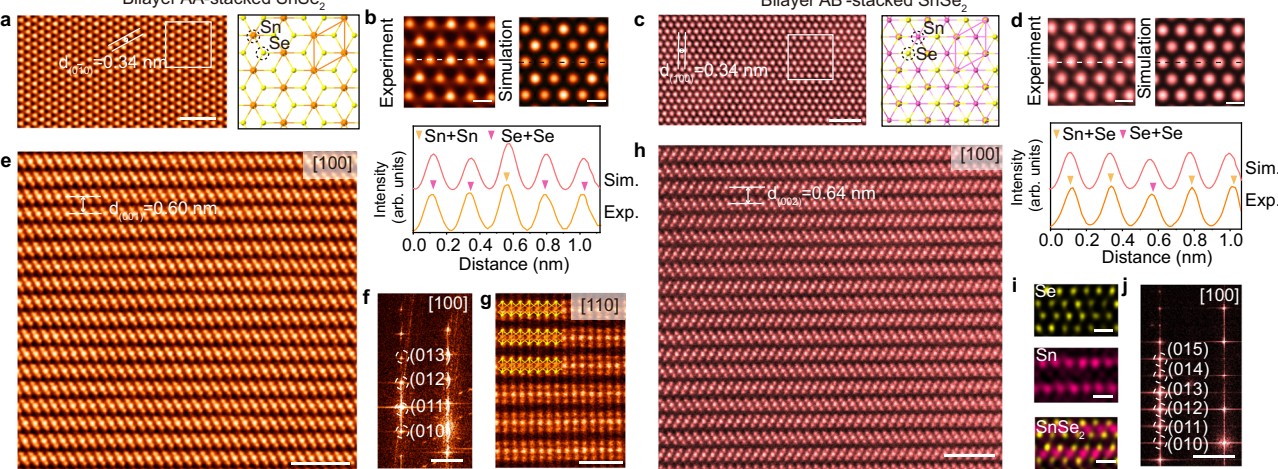

**Fig. 2 | Atomic-resolution images of AA- and AB'-stacked SnSe₂.** **a** Annular dark field scanning transmission electron microscopy (ADF-STEM) image of AA-stacked SnSe₂ along the [001] direction. The corresponding atomic model was depicted on the right panel. **b** Enlarged ADF image from a with simulated image calculated from AA-stacked SnSe₂. Intensity line profiles derived from the experiment and simulation were shown in the lower panel. **c** ADF-STEM image of AB'-stacked SnSe₂ along the [001] direction. The atomic model was depicted on the right panel. **d** Enlarged ADF image from c and corresponding simulated image derived from AB'-stacked SnSe₂. Intensity line profiles extracted from the experiment and simulation were depicted in the lower panel. **e** Cross-sectional ADF-STEM image of AA-stacked SnSe₂ along the [100] zone axis and **f** its corresponding fast Fourier transformation (FFT) pattern. **g** Cross-sectional ADF-STEM image of AA-stacked SnSe₂ along the [1̄10] zone axis. **h** ADF-STEM image of AB'-stacked SnSe₂ along the [100] direction. **i** Corresponding EDS mapping showing Se, Sn, and overlaid image. **j** The FFT pattern derived from h. Scale bars: **a**, **c** 1 nm, **b**, **d** 0.2 nm; **e**, **h** 2 nm; **f**, **j** 2 nm⁻¹; **g** 1 nm; **i** 0.5 nm.

due to their inherent centrosymmetry unless spontaneously broken centrosymmetry through interlayer gliding or inversion symmetry operations. Notably, the crystals grown at 320 °C show no SHG emission, indicating a centrosymmetric AA stacking arrangement. On the other hand, crystals synthesized at 340 °C demonstrate a robust SHG signal implying a unique interlayer stacking configuration with a broken centrosymmetry (Fig. 1e and Supplementary Figs. 7–11).

To atomically elucidate the atomic structure and stacking sequences of two distinct SnSe₂ crystals, we have carried out annular dark field scanning transmission electron microscopy (ADF-STEM) imaging. It is known that the intensity of the atom blobs in the ADF image is nearly proportional to the atomic number $Z^{1.6-1.7}$, which has been widely employed to atomically visualize the atomic structure and stacking polytypes of 2D materials[48]. The ADF-STEM images (Fig. 2a) of flakes grown at 340 °C reveal that they are highly crystalline and free of topological defects. The projected Z-contrast pattern (Fig. 2b) greatly resembles the AA-stacked SnSe₂ as corroborated by the image simulation (Fig. 2b and Supplementary Fig. 12a–c), where the two dim atom blobs correspond to the Se (Z = 34) dimer sites, and the bright atom

blobs are composed of pure Sn (Z = 50) atoms. On the other hand, the contrast pattern is distinct in crystals grown at 340 °C. Notably, there are two bright atom blobs and one dim atom blob in each unit cell. This configuration occurs due to interlayer sliding, which alters the intrinsic overlap of Se and Se atoms in bilayers being altered to an overlap of Se and Sn atoms, consequently reducing atomic brightness. The contrast disparity between bright and dim atom blobs is largely reduced compared to the ones in AA-stacked SnSe₂ (Fig. 2c). This unique contrast pattern is due to the interlayer AB' stacking registry having a $P6_3mc$ space group, in which one SnSe₂ slab undergoes a mirror symmetry operation plus an interlayer gliding along the [120] direction for $\frac{1}{\sqrt{3}}$ a against the other, as further verified by the image simulation (Fig. 2d and Supplementary Fig. 12d–f).

To accurately discern the atomic-scale stacking sequences of SnSe₂, we conducted cross-sectional ADF-STEM imaging. All samples were carefully prepared and cleaned by the focused ion beam (FIB) with low acceleration voltage to largely dissipate the ion damage. Notably, a perfect AA stacking polytype without any structural distortions along the [100] zone axis was observed (Fig. 2e) agreeing well

with the top-view results. The corresponding fast Fourier transformation (FFT) pattern (Fig. 2f) exhibits a single set of spots, indicative of a perfect AA stacking order (Supplementary Fig. 13a, b, f, g). The ADF-STEM images captured along the [1$\bar{1}$0] direction (Fig. 2g) further reveal a well-defined AA stacking sequence. In parallel, an ordered AB′ stacking polytype was verified by cross-section ADF-STEM imaging (Fig. 2h), in which every SnSe$_2$ slab alternatively shifts laterally by $\frac{1}{\sqrt{3}}a$ unit cell and rotates 180° against the other. Atomic-scale EDS mapping (Fig. 2i) confirmed that the AB′-stacked SnSe$_2$ crystals are free of antisite defects or impurities. Because of the long-range ordering (more than 25 nm) in AB′-stacked SnSe$_2$ along the [1$\bar{1}$0] direction, we found doubling superspots in the corresponding FFT patterns (Fig. 2j and Supplementary Fig. 13c−e, h−j). To confirm the large-scale consistency of the stacking sequence, we performed low-magnification ADF-STEM imaging along with corresponding FFT pattern analysis. These results clearly demonstrate the successful synthesis of pure AA- and unconventional AB′-stacked SnSe$_2$ crystals (Supplementary Fig. 14). Based on the demonstration of AB′-stacked SnSe$_2$, we conducted piezoresponse force microscopy (PFM). As illustrated in Supplementary Fig. 15, the observed phase hysteresis and butterfly-shaped amplitude loops are an indication of polarization switching, demonstrating the presence of out-of-plane ferroelectricity in AB′-stacked SnSe$_2$ crystals.

## Demonstration of substrate-guided growth mechanism

Next, we atomically unveil the underlying growth mechanism of unique AB′-stacked SnSe$_2$. First, we statistically analyze the epitaxial relationship between the SnSe$_2$ crystal orientation and the mica substrate to examine the substrate effect. After surveying a few hundred flakes, we found that (Fig. 3a) the as-grown triangular SnSe$_2$ domains predominantly manifest two distinct orientations (0° and 30°) with regards to the mica substrate, corresponding to 96% (0°) and 4% (30°) over the total population, respectively, validating the strong epitaxial relationship between the mica substrate and SnSe$_2$ crystals (Fig. 3b, c and Supplementary Figs. 16, 17). To precisely unveil the epitaxial relationship at an atomic level, we performed atomic-resolution ADF-STEM imaging along the interface. Notably, the cross-sectional ADF-STEM images along the [1$\bar{1}$0] direction confirmed an obvious lattice-matching relationship. Both AA- and AB′-stacked SnSe$_2$ display the same lattice constant of 3.87 Å, substantially away from the lattice constant (5.31 Å) of the mica substrate (Fig. 3d). The FFT pattern derived from the ADF-STEM image indicates that the (010) crystallographic plane of SnSe$_2$ aligns with the (100) crystal plane of mica (Fig. 3e, f). The pattern thus corresponds to a nearly commensurate superlattice comprising 4 × 4 SnSe$_2$ unit cells (4 × 3.87 = 15.48 Å) and 3 × 3 mica cells (3 × 5.31 = 15.93 Å). The remaining 2.8% tensile strain could be gradually accommodated by a few SnSe$_2$ unit lengths adhered to the mica substrate. Furthermore, we observed a certain degree of ripples at the interface, as labeled by a white dashed line, indicating that interfacial charge transfer is potentially responsible for interlayer distortions (Fig. 3e).

To further elucidate the underlying substrate-guided growth mechanisms, we performed density functional theory (DFT) calculations (Supplementary Note 1). Initially, the formation energy of AA, AB′, AB, and AA′ four stacking registries was assessed (Supplementary Fig. 18 and Supplementary Table 1). Taking AA-stacked SnSe$_2$ as the

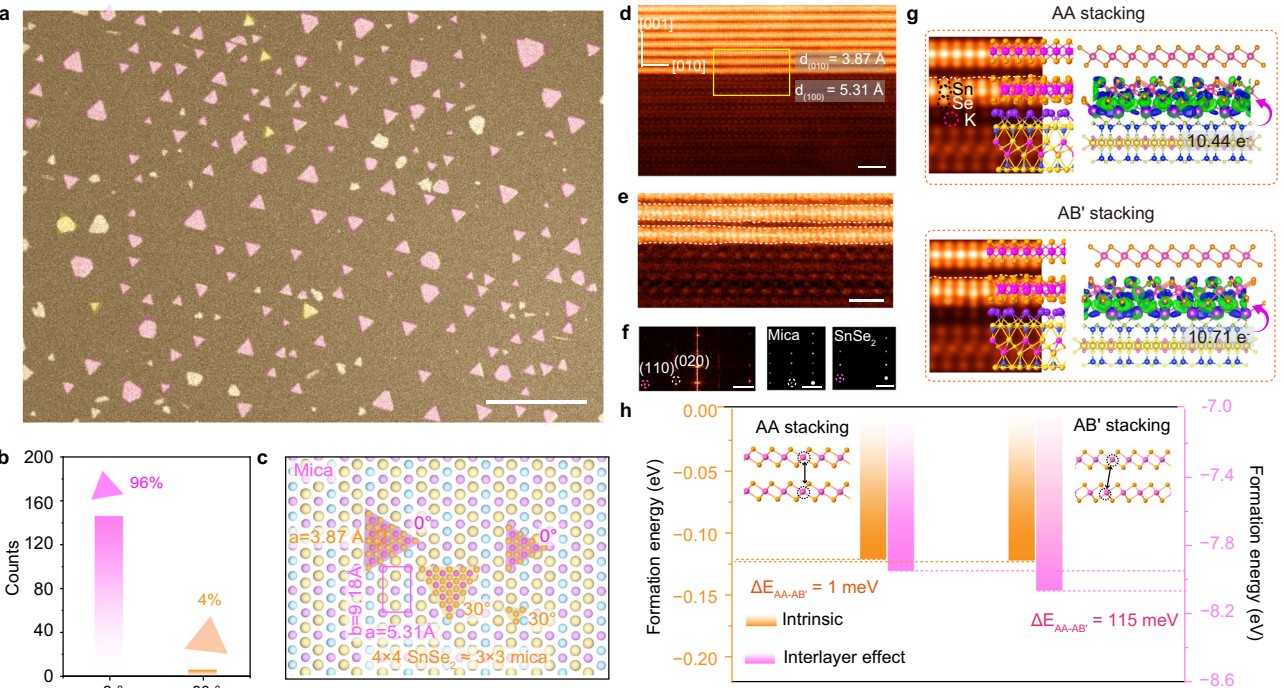

**Fig. 3 | Underlying mechanisms for growing SnSe$_2$ with novel phases.**
**a** Scanning electron microscopy (SEM) image showing epitaxial growth of SnSe$_2$ flakes on mica substrate. **b** The statistical counting shows the distribution of 0° and 30° twisted SnSe$_2$ flakes grown on mica. **c** Schematic illustration depicting SnSe$_2$ flakes grown on mica substrate with different crystallographic orientations. **d** The atomic-resolution cross-sectional ADF-STEM image of AB′-stacked SnSe$_2$/mica substrate along the [1$\bar{1}$0] zone axis. **e** The zoom-in ADF-STEM image reveals atomic distortion as depicted by the white dashed lines along the interface.
**f** Corresponding FFT pattern from (**e**). Simulated FFT patterns of mica and AB′-stacking SnSe$_2$ were shown on the right panels. **g** The simulated ADF-STEM

images derived from the density functional theory (DFT)-optimized AA- and AB′-stacked SnSe$_2$ on the mica surface. The right panel showing the differential charge density of AA-, and AB′-stacked SnSe$_2$ on mica substrate. The area in green is the region that gained electrons, and the area in blue is the region that lost electrons. **h** DFT calculated formation energies of AA-stacked SnSe$_2$ and AB′-stacked SnSe$_2$ before and after considering the interface effect, respectively. $\Delta E_{AA-AB'}$ denotes the difference in formation energy. The orange dashed line represents the formation energy of AA stacking, while the pink dashed line represents the formation energy of AB′ stacking, both represent with and without the influence of the mica substrate, respectively. Scale bars: **a** 10 μm; **b**, **e** 1 nm; **f** 2 nm$^{-1}$.

ground state, both AA and AB' stacking sequences are stable, differing by merely 1 meV/formula unit. Conversely, AA' and AB polytypes are thermodynamically unstable, exhibiting additional formation energies of 11.5 and 14 meV, respectively (Supplementary Fig. 19) compared to AA stacking registry. Notably, mica substrates, characterized by their layered aluminosilicate structure owning surface potassium ions for charge balance, facilitate charge transfer at interfaces with 2D materials and enhance interfacial interactions (Supplementary Fig. 20). Differential charge density calculations suggest that there is substantial charge transfer along the interface, with 10.71 e⁻ being transferred from the K atoms to bridging Se atoms in AB'-stacked $SnSe_2$, and 10.44 e⁻ in AA-stacked $SnSe_2$ (Fig. 3g and Supplementary Table 2).

The charge transfer significantly reduces the formation energy of AB'-stacked $SnSe_2$ by 115 meV, which has improved the possibility of controlled synthesis for two stacking polytypes (Supplementary Table 3). Calculations indicate that the formation of AB'-stacked $SnSe_2$ is attributed to receiving additional electrons supplied by the mica substrates (Supplementary Fig. 21). Thus, the AA and AB' stacking sequences are believed to be predominantly triggered via the synergistic effect of both temperature thermodynamics and substrate-guided processes (Fig. 3h)[49]. On the contrary, pure AB' stacking was not observed on sapphire substrates (Supplementary Figs. 22–26). Upon characterizing the stacking behaviors, we found that 75% exhibited BA' stacking behaviors, 3.6% with AB' stacking orders, and 21.4% had intrinsic stacking slabs. The weakened charge transfer process between sapphire and $SnSe_2$ crystals results in only a 28 meV difference in formation energy difference, highlighting the critical role

of the substrate (Supplementary Fig. 27, Supplementary Tables 4, 5, and Supplementary Note 2).

## Minor interlayer gliding in AB'-stacked $SnSe_2$ crystals

The as-grown AB'-stacked $SnSe_2$ polytype, with a non-centrosymmetric $P6_3mc$ space group, exhibits potential for NLO responses. To verify the hypothesis, we conducted linearly polarization-dependent SHG intensity measurements of AB'-stacked $SnSe_2$ crystals. Unlike an expected 6-lobe anisotropic SHG polar plot in 3-fold rotational symmetry in AB'-$SnSe_2$, we observe an almost azimuthal-polarization dependence of the SHG signal (Fig. 4a and Supplementary Fig. 28). This might be attributed to minor interlayer gliding of $SnSe_2$ slabs along the armchair or zigzag direction (Fig. 4b), reducing the intrinsic in-plane sixfold rotational symmetry to a nearly twofold. Therefore, we calculate the polarization-dependent SHG signals by introducing trivial interlayer gliding along the armchair direction for 0.17 Å (Supplementary Fig. 29). To atomically unveil the picometer scale interlayer gliding, we developed customized coding to visualize the degree of the gliding (λ) and successfully detected trivial interlayer sliding (Fig. 4c–g, Supplementary Figs. 30, 31, Supplementary Note 3, and Supplementary Table 6). The minor interlayer gliding is ubiquitous due to the negligible gliding barrier in 2D materials, which can be readily overcome by thermal scattering and other kinetic factors[50]. As suggested by DFT calculation (Fig. 4h and Supplementary Table 7), the gliding barrier is only 1.54 meV/f.u. if the gliding magnitude is less than 0.22 Å along [120] direction in AB'-stacked $SnSe_2$. The subtle gliding significantly alters the materials' optical properties, as evidenced by the observed variation in

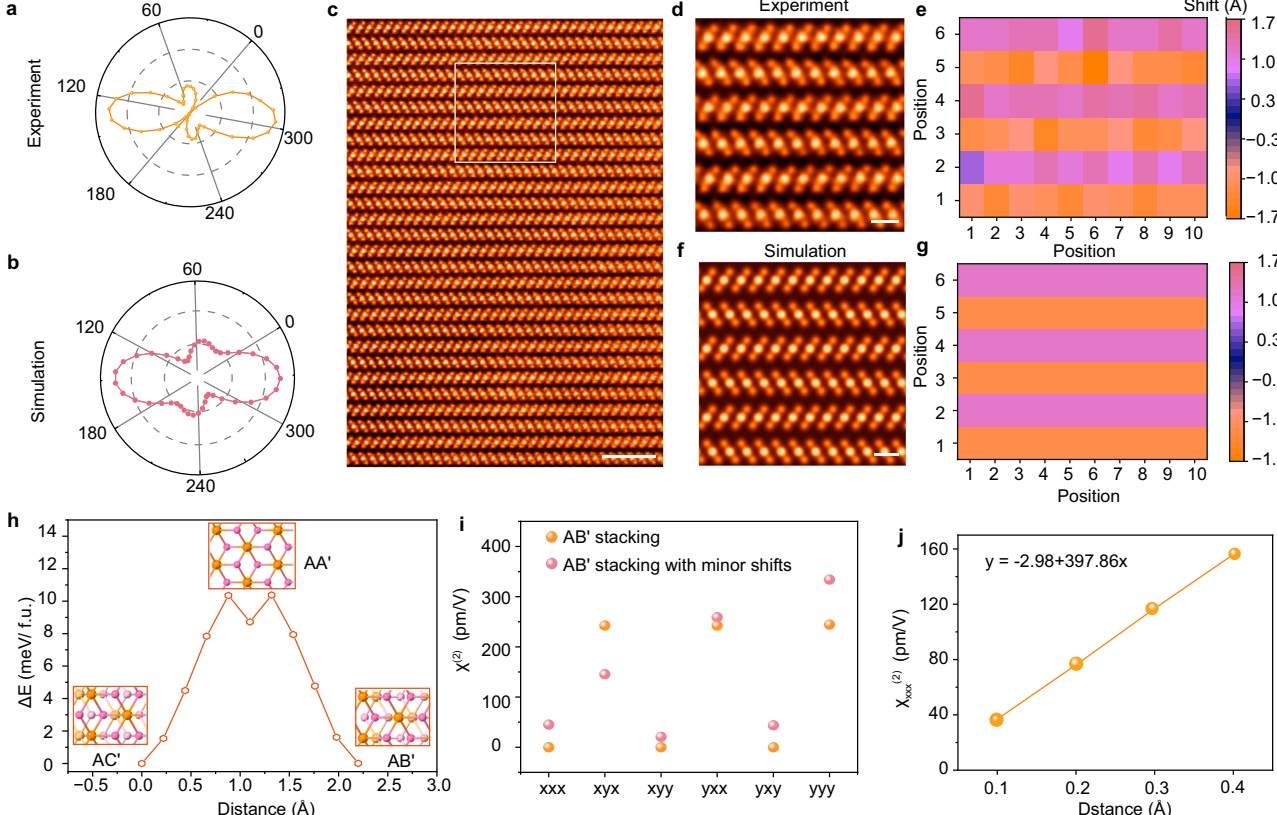

**Fig. 4 | Atomic structure analysis and associated spontaneous SHG. a** The polarization-resolved SHG response with a 1064 nm excitation laser. **b** Calculated SHG response derived from the AB' stacking atomic model containing 0.17 Å interlayer gliding along the [120] direction. **c** Cross-sectional ADF-STEM image exhibiting a long-range order AB'-stacked $SnSe_2$ along the [100] zone axis. **d** Enlarged ADF-STEM image marked by the white box in (**c**) and corresponding Sn atom displacement map (**e**). **f** Simulated ADF-STEM image of pristine AB'-stacked $SnSe_2$ and corresponding Sn atom displacement map (**g**). **h** DFT calculated energy landscape gliding from the AC' stacking order to the AB' stacking order using a bilayer $SnSe_2$ model along the [120] direction. **i** The nonlinear susceptibility tensor element $\chi^{(2)}$ of AB'-stacked $SnSe_2$ with minor sliding (orange) and pure AB' stacking sequence (pink). **j** The $\chi^{(2)}_{xxx}$ component as a function of interlayer gliding along the [120] direction. Scale bars: **c** 2 nm; **d**, **f** 0.5 nm.

the frequency-dependent $\chi^{(2)}$ (Fig. 4i). Notably, six nonvanishing $\chi^{(2)}$ tensors are observed in contrast to AB'-stacked $SnSe_2$, which exhibits only three nonzero $\chi^{(2)}$ due to the breaking of 3-fold rotational symmetry by interlayer gliding (Fig. 4h and Supplementary Fig. 29b).

We further performed the first-principle calculations on structures with interlayer gliding along the zigzag direction in AB'-stacked $SnSe_2$. The resulting SHG polar plots, and the $\chi^{(2)}$ components (Supplementary Fig. 32) are robust against the interlayer shift and highly sensitive to local distortion. By adjusting the interlayer gliding to increments as small as 0.1 Å, numerical calculations establish the $\chi_{xxx}^{(2)}$ diagram as a function of shift magnitude, showing a linear dependence of the magnitude of SHG susceptibility $|\chi_{xxx}^{(2)}|$ on the gliding distance, which results into a quadratic relationship between SHG signal intensity and shift magnitude (Fig. 4j). To this end, a causal link between interlayer gliding behaviors and NLO response is established. Specifically, a series of NLO responses can be regulated by introducing the interlayer gliding.

## The synthesis of high-order SnSe₂ stacking polytypes

It has been already demonstrated that one periodic symmetry operation could confer rich NLO responses. Next, we systematically unravel a spectrum of NLO variations upon precisely interlayer symmetry operations in $SnSe_2$. It is known that 1T-$SnSe_2$ is a threefold symmetry, where the Sn layer is sandwiched between two Se layers with Se-Sn-Se atomic stacking. The crystallographic orientation could be classified into two degenerate directions (Fig. 5a), e.g., zigzag directions are along with the basic vectors [100] and [010] directions, armchair directions along [120] (pink arrows). Three high-symmetry sites are indexed as A (Sn atoms), B (top Se atoms), and C (bottom Se atoms), and the distinct AA, AB, and AC stacking orders can be formed via the gliding of a $SnSe_2$ monolayer along the armchair direction for 0, $\frac{1}{\sqrt{3}}a$, and $\frac{2}{\sqrt{3}}a$, respectively (Fig. 5b). The AB' polytype can be regarded as imposing a periodic in-plane interlayer gliding along the [120] direction for $\frac{1}{\sqrt{3}}a$ plus an inversion symmetry operation against the bottom layer, in which we denoted the interlayer symmetry operation as R'. In parallel, there exist R, L, and L' interlayer symmetry operations as illustrated in Fig. 5b. In our experiments, we found that each $SnSe_2$ slabs undergo R', L', or their combinational operations, rendering various types of high-order superlattices.

In our experiments, we identified four distinct high-order stacking polytypes: (1) the 6 R ($R\bar{3}m$) phase, (2) the 18 R ($R\bar{3}m$) stacking, (3) the 12 R ($R3m$) stacking sequence, (4) the 18 C ($Cm$) monoclinic phase, which are formed by R', R'R'L', R'R'R'L', and R'R'R'R'R'L' periodic symmetry operations, respectively, as verified by cross-sectional ADF-STEM images (Fig. 5c–f and Supplementary Fig. 33). The numerical values denote the layers per repeating unit in $SnSe_2$, while the letters R and C designate the respective crystal symmetries[51]. The aforementioned AB' stacking polytype is achieved via periodic R' and L' symmetry operations. All $SnSe_2$ samples demonstrate that the mirror-symmetric pairs of the tilted $SnSe_2$ slabs slide alternatively to the right [120] or left [$\bar{1}$20] orientations, leading to the binding of different unit cells with distinct crystal symmetries (Fig. 5g–j), as highlighted by pink and green boxes. The resulting higher-order symmetries are verified by the appearance of additional diffuse streaks in the corresponding FFT patterns, consisting of the emulated patterns[52] (Supplementary Fig. 34).

To unveil the underlying growth mechanism of different degrees of $SnSe_2$ superlattices, we quantified the interlayer energies of various symmetry operations. It can be seen that the interlayer energies of R' and L' operations are observed to be the most thermodynamically favored stacking orders as compared to R and L operations by 14 and 43 meV, respectively (Supplementary Fig. 35). When the mica substrate is taken into account, electron transfer from the mica to the $SnSe_2$ crystals disrupts the energy degeneracy, resulting in an interlayer energy difference of 0.601 eV. These results lead to the stabilization of higher-order $SnSe_2$ stacking polytypes (Fig. 5k and Supplementary Table. 8). In our experiments, various stacking orders can be

effectively synthesized within a narrow sublimation temperature range of 320–340 °C. Calculations imply that CVD-grown $SnSe_2$ with diverse layer gliding behaviors are highly challenging owing to these narrowly defined growth conditions. For a comprehensive understanding of the growth dynamics, refer to Supplementary Note 4, and the detailed growth conditions based on modulating the gas flow in a range of 40–80 sccm to grow various stacking orders are listed in Supplementary Table 9. Excitingly, the proposed kinetic-guided strategy can be extended to synthesize other TMDs with distinct crystal symmetry. Supplementary Fig. 36 shows that $TiSe_2$ displays an alternative in-plane inversion asymmetry with a stacking sequence of AAC'C'BBC'C'...., significantly distinct from the well-documented AA stacking sequence.

Finally, we systematically discuss the stacking-dependent strength of SHG response. The detailed analysis presented in Supplementary Note 5 and Supplementary Table 10, reveals the variations in SHG responses based on different stacked $SnSe_2$. As shown in Fig. 5l, m, the calculated $\chi^{(2)}$ of $R\bar{3}m$ space group is weak, consistent with the space group featuring a threefold rotoinversion axis. Conversely, the $\chi^{(2)}$ tensors of $R3m$ stacking polytype, which pertains to a non-centrosymmetry system, demonstrated a noticeable enhancement at an incident excitation of 1064 nm (Fig. 5n). The interlayer gliding diversifies the crystal symmetry and stacking sequences, paving the way for effective property modulations, such as ferroelectric[53], ferromagnetism[32,36,54] and catalytic[55]. The modulation of centrosymmetry via interlayer gliding presents great possibilities for broadening the scope of two-dimensional ferroelectric materials. Despite the intrinsic neutrality and identical composition in each layer, their asymmetrical charge density distribution, resulting from the interlayer configuration, leads to vertical polarity. These observations are further substantiated by theoretical calculations based on SHG measurement. In addition, it is noteworthy that additional optical anisotropic characterization, such as polarized Raman spectroscopy on AB'-stacked $SnSe_2$ crystals, and the results are presented in Supplementary Note 6.

In summary, our proposed substrate-guided growth strategy has been demonstrated to be effective in the synthesis of a wide range of $SnSe_2$ superlattices with $P\bar{3}m1$, $P63mc$, $R\bar{3}m$, $R3m$, $Cm$ etc., phases. The interlayer symmetry operations can be well controlled into single-atomic-plane precision. The substrate guides the initial atomic arrangements of crystals and provides the momentum for subsequent $SnSe_2$ slabs sliding and/or inversion. According to first-principle calculations, crystal symmetry-dependent NLO responses have been systematically established in an atlas of higher-order $SnSe_2$ superlattices. We further demonstrated that such substrate-guided grown superlattices with atomic-controlled interlayer symmetry operations can be also achieved in other 2D materials. This work brings the inspiration for the controllable growth of 2D materials with precise crystal symmetry, paving the way for tunning topological trivial polytypes with single-atomic-scale precision, and shed light on new quantum phenomena triggered by stackingtronics.

## Methods
### Synthesis of various polytypes of SnSe₂
$SnSe_2$ flakes were synthesized on fluorophlogopite mica ($KMg_3AlSi_3O_{10}F_2$) substrate via ambient CVD method. The reaction was conducted in a single-zone furnace equipped with a one-inch quartz chamber. -20 mg $SnI_2$ powder (Alfa Aesar, 99.999%) and -100 mg Se powder (Alfa Aesar, 99 + %) were loaded in an alumina boat upstream. Freshly cleaved mica (Changchun Fluorphlogopite Mica Company Ltd, $10 \times 10 \times 0.2$ mm) put 15 cm away from reaction precursors. Prior to the reaction, the furnace was washed with 200 sccm high-purity Ar gas for 30 min to remove the residual gas. Then, 20 sccm Ar and 0.8 sccm $H_2$ were optimized to create the growth environment. The growth zone was heated to 600 °C ($SnI_2 + 2Se + H_2 = SnSe_2 + 2HI$) for 20 min. Subsequently, $SnI_2$ precursors and Se powers were rapidly pushed into the heating zone and kept for another 5 min. By adjusting various gas flows

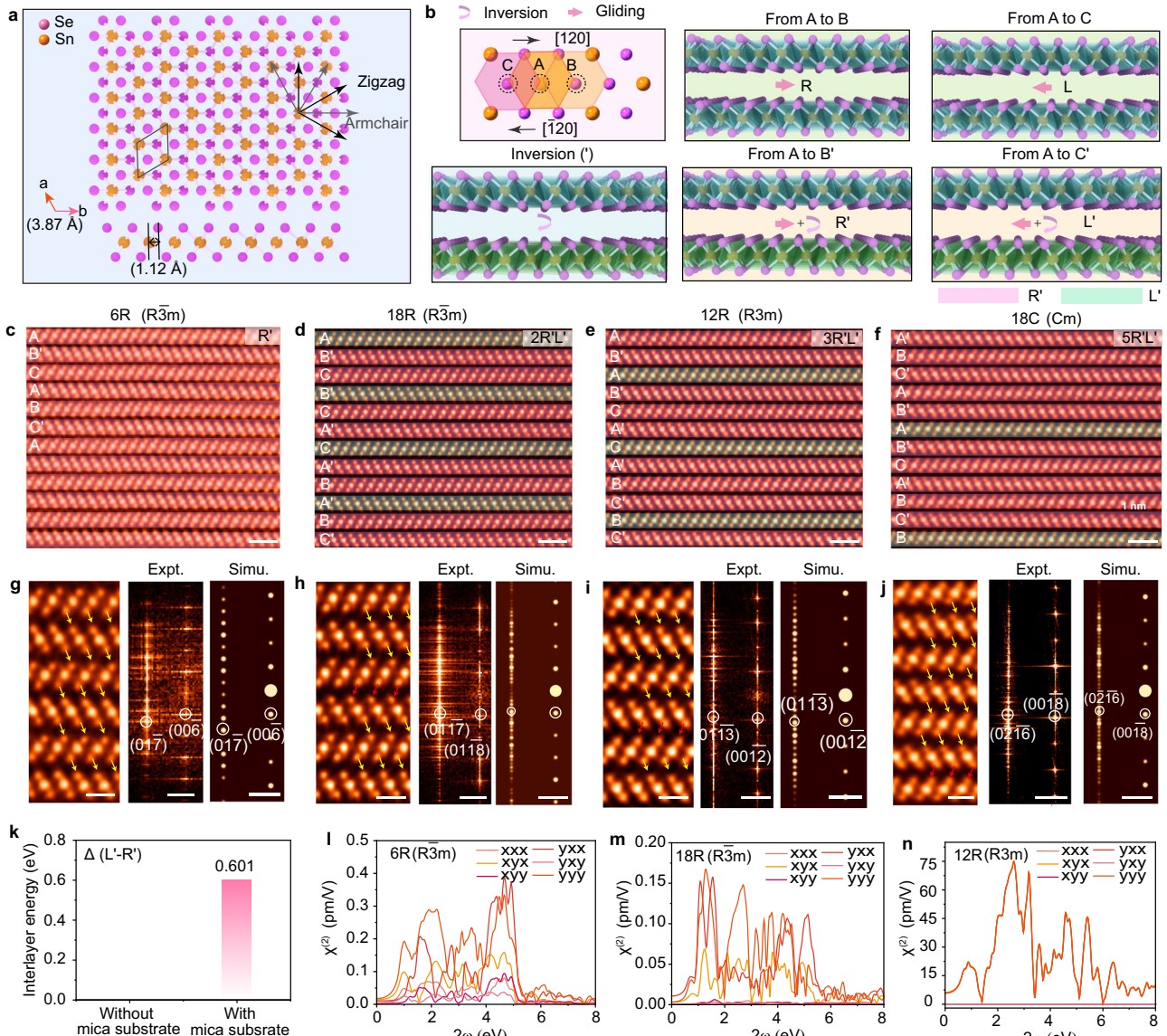

**Fig. 5 | A spectrum of stacking polytypes induced by a variety of interlayer symmetry operations in SnSe₂. a** Top and side views of monolayer SnSe₂. The armchair [120] and zigzag [110] directions are marked with the gray arrow and black arrow, respectively. **b** Atomic models of bilayer SnSe₂ showing five distinct symmetry operations in sequence, i.e., gliding $\frac{1}{\sqrt{3}}$ a from A to B defined as R, gliding $\frac{1}{\sqrt{3}}$ a from A to C defined as L, interlayer gliding from A to B combining with inversion named as R′, interlayer gliding from A to C combining with inversion behaviors named as L′. Cross-sectional ADF-STEM images of SnSe₂ superlattices showing 6 R ($R\bar{3}m$) (**c**), 18 R ($R\bar{3}m$) (**d**), 18 R ($R3m$) (**e**), and 18 ($Cm$) (**f**) space groups. The red and yellow false colors were used to label the gliding direction towards the right-hand side direction [120] and left-hand side [$\bar{1}$20], respectively. **g–j** Enlarged ADF-STEM images extracted from (**c–f**) presenting the unique stacking orders, respectively. Corresponding FFT patterns and simulated FFT patterns were depicted on the right panels. **k** Calculated interlayer energies of R′ and L′ operation symmetry in (**b**). Theoretical calculated $\chi^{(2)}$ elements of 6 R ($R\bar{3}m$) (**l**), 18 R ($R\bar{3}m$) (**m**), 18 R ($R3m$) (**n**) stacked SnSe₂. Scale bars: **c–f** 1 nm; **g–j** Left, 0.5 nm; Right, 2 nm⁻¹.

in a range of 40–80 sccm, we intentionally synthesized various SnSe₂ stacking sequences.

## Sample characterization and image simulation

Optical images were characterized by a Nikon microscope. AFM (Bruker, Dimension Icon) was employed to measure thicknesses. PFM measurements were conducted on flakes exfoliated onto Au film deposited on Si substrate, using a probe with a ~3 N/m spring constant and a conductive Pt/Ir coating layer. Out-of-plane PFM signal was recorded utilizing a driving frequency of 383 kHz and a drive amplitude of 1500 mV. A bidirectional bias sweep between −8 and 8 V was applied to measure the hysteresis loop. Low-magnification EDS mapping (FEI, Tecnai F20) was used to analyze element compositions.

Raman spectra were collected by Witec with an excitation light of ~532 nm. The atomic structures of SnSe₂ flakes were characterized by a cold-field emission transmission electron microscope (Titan Cubed Themis G2 200) operating at 300 kV. The collection angle of the ADF images ranges from 80 to 200 mrad. Image simulations were performed with the Prismatic package, assuming an aberration-free probe with a probe size of ~1 Å.

## Data availability

The Source Data underlying the figures of this study are available with the paper. All raw data generated during the current study are available from the corresponding authors upon request. Source data are provided with this paper.

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

## Acknowledgements

X.Z. thanks the National Key R&D Program of China (2024YFE0109200), the Beijing Natural Science Foundation (JQ24010, Z220020), the National Natural Science Foundation of China (Grant No.52273279), and the open research fund of Songshan Lake Materials Laboratory (Grant No.2023SLABFN26). D.G. acknowledges the financial support from the National Key R&D Program of China (2022YFC3401200 and 2021YFA0717900), the Natural Science Foundation of Tianjin (Grants 22JCJQJC00080); the Natural Science Foundation of China (Grant 52472168; the Beijing National Laboratory for Molecular Sciences (BNLMS202309); the Fundamental Research Funds for the Central Universities. J.Z. thanks the Ministry of Science and Technology of China (2022YFA1203302, 2022YFA1203304, and 2018YFA0703502), the National Natural Science Foundation of China (Grant Nos. 52021006), the Strategic Priority Research Program of CAS (XDB36030100), the Beijing National Laboratory for Molecular Sciences (BNLMS-CXTD-202001), and the Shenzhen Science and Technology Innovation Commission (KQTD20221101115627004). L.W. acknowledges the support from the National Natural Science Foundation of China (Grant No. 52302187). The authors acknowledge the Electron Microscopy Laboratory of Peking University, China, for the use of Cs-corrected Titan Cubed Themis G2 200 transmission electron microscopy. The authors thank the Materials Processing and Analysis Center, Peking University, for assistance with TEM, SEM, and Raman characterization.

## Author contributions

X.Z., D.G., and W.H. conceived and designed the experiments. Z.H. synthesized materials and conducted SHG, Raman, and XRD characterizations. Z.H. and S.W. performed the FIB and ADF-STEM characterizations. F.X. conducted the calculations of nonlinear optical properties for the SnSe₂. C.H. and J.Q. conducted the DFT calculations. J.L., Y.M., and J.Z. conducted the identification of atomic positions. L.W. and Y.L. helped with the PFM characterizations. Z.H. wrote the manuscript and X.Z., Q.Z., and X.H. polished the manuscript. All the authors discussed the results and contributed to preparing the manuscript.

## Competing interests

The authors declare no competing interests.
