## [Transparent Peer Review file · Nature Communications]

Atomically Engineering Interlayer Symmetry Operations of Two-Dimensional Crystals

Corresponding Author: Professor Xiaoxu Zhao

Version 0:

Reviewer comments:

Reviewer #1

(Remarks to the Author)

In the manuscript titled "Atomically Engineering Interlayer Symmetry Operations of Two-Dimensional Crystals", the authors present a CVD growth method to grow AB'-stacked SnSe₂. Also, they found many phases like 6R, 12R, 18R, 18C, etc., in SnSe₂. The phase controlled growth of 2D materials is very important and interesting. However, the data presented in the current version didn't represent a substantial methodological advance over previous reports to support its publication. My specific comments are as follows:

1. Many works have reported the controllable CVD growth of few layer SnSe₂ on mica, such as Adv. Mater. 27, 8035–8041 (2015), Adv. Mater. Interfaces 9, 2102376 (2022) and RSC Adv. 10, 42157–42163 (2020). Some of them also reported the controlled-phase growth of SnSe₂. According to the data presented at present, the growth method is very similar to the previous reports.
 2. The authors were only able to achieve partial AA and AB'-stacked SnSe₂ on one substrate, and this work would be of greater significance if complete AB'-stacked SnSe₂ growth could be attained.
 3. Following the previous question, although the authors have presented several interesting phases through TEM data, there is a lack of discussion on the controllability and occurrence probability of these phases like 6R, 12R, 18R, 18C, etc., in SnSe₂. If these phases can only be observed in the sample without being reproducibly prepared over a large area, their significance would be significantly diminished. Additionally, in the method section, only changing gas flow can obtain different SnSe₂ stacking, the underlying mechanism remains incomprehensible.
 4. The calibration of the current orientations of the SnSe₂ islands has some uncertainty. In Figure 2, the author employs red and yellow dotted lines to represent two orientations; however, it is evident that the edges of many triangles significantly deviate from the provided yellow dotted lines, indicating an estimated deflection of approximately 10 degrees.
- In general, I strongly concur with the author's perspective that symmetry regulation plays a crucial role in two-dimensional materials. However, the data provided by the authors in the current version does not substantiate their ability to effectively synthesize two-dimensional materials with controllable phases.

Reviewer #2

(Remarks to the Author)

This manuscript presents a significant advancement in the field of 2D materials and crystal engineering. The authors have successfully demonstrated a novel method for controlling interlayer symmetry operations, which has potential implications for a wide range of applications. The study is well-executed, and the findings are supported by robust experimental and theoretical evidence. However, some sections could benefit from more concise writing to improve readability. Overall, the manuscript makes a valuable contribution to the field and is suitable for publication after minor revisions for clarity and conciseness.

1. Is the interlayer sliding mentioned in the article controllable? Are the different high-order symmetries randomly generated?
2. What were the reference materials or benchmarks used to validate the SHG measurements?
3. Why Mica substrate was used in the study and other substrates (e.g. sapphire, MgO) are not available?
4. Although the research results indicate that different stacking sequences and symmetries affect the second harmonic generation (SHG) signal, most of the findings rely on theoretical calculations and simulations. Experimental validation is primarily focused on a limited number of samples, lacking experimental data support under broader conditions and materials.

5. In addition to the optical anisotropy characterizations mentioned in the article, other optical anisotropy characterizations (e.g., Raman spectroscopy) can be added to the Supplementary Note.

Reviewer #3

(Remarks to the Author)

In this work, Han et al. studied SnSe₂ grown on mica by a CVD method that utilizes SnI₂ and Se as the precursors. They proposed a growth mechanism to explain how potassium terminated on mica could stabilize the metastable AB'-stacked SnSe₂ through the interlayer charge transfer. In the second part of the manuscript, the authors focused on the atomic analysis of ADF-STEM images of AB'-SnSe₂ multilayers, including some of the stacking polytypes and their NLO properties. While a lot of interesting data were presented and the results are relatively new compared to similar SnSe₂ work, I don't fully agree with their conclusions. The mica they used indeed stabilized some of the SnSe₂ domains into AB'-stacked layers and helped epitaxy. However, I don't think it can guide SnSe₂ to grow into a particular stacking polytype. Fig.2 shows the distribution of 0- and 30-degree epi-SnSe₂ on mica. However, the authors did not correlate how many of the 0 degrees and how many of the 30 degrees are AB' stacking. I think that AB' type flakes must have a unique epitaxial relationship with mica to have the interlayer effect discussed in Fig.2g. (Perhaps the authors can come out with a rapid SHG characterization method to examine a number of domains orienting at both 0- and 30-degree on mica.) Moving forward to Fig.4, the authors found several stacking polytypes in ADF-STEM experiments and used interlayer sliding, gliding, and inversion to explain their formation. While these stacking polytypes are certainly interesting and provide great insights into the community, the authors cannot link their formation to the substrate-guided growth they proposed. I also think it would be great if the authors linked these polytypes to any ferroelectric properties or showed some examples of property modulation. To make their formation controllable, layer number control capabilities and chemical potential balance between Sn and Se need to be established, such as by MOCVD [e.g., Adv. Mater. 2024, 36, 2400800, in which SnSe₂ that grows at 200 oC was reported]. Based on my observation, I feel that this work might be more suitable for other 2D material journals or should be returned to the authors for a major revision.

Version 1:

Reviewer comments:

Reviewer #1

(Remarks to the Author)

The authors have made great efforts and conducted additional experiments to address my comments. The current version has undergone substantial enhancements, rendering it suitable for publication now.

Reviewer #2

(Remarks to the Author)

The revision successfully covers all my concerns and the manuscript is recommend to publish in Nature Communication.

Reviewer #3

(Remarks to the Author)

I carefully read their answers and felt that they addressed the questions I raised on their 1st submission well. I have no further questions.

Reviewer #1

In the manuscript titled "Atomically Engineering Interlayer Symmetry Operations of Two-Dimensional Crystals", the authors present a CVD growth method to grow AB'-stacked SnSe₂. Also, they found many phases like 6R, 12R, 18R, 18C, etc., in SnSe₂. The phase controlled growth of 2D materials is very important and interesting. However, the data presented in the current version didn't represent a substantial methodological advance over previous reports to support its publication. My specific comments are as follows:

Reply: We appreciated the reviewer for the careful review and constructive suggestions. Despite the common CVD method adopted in this manuscript, we have carefully analyzed and elucidated the critical role of the substrate effect in modulating the high-order phases. In addition, we have conducted additional experiments to further unveil the controllability of the synthesis process as appended in the following answers.

1. Many works have reported the controllable CVD growth of few layer SnSe₂ on mica, such as *Adv. Mater.* 27, 8035-8041 (2015), *Adv. Mater. Interfaces* 9, 2102376 (2022) and *RSC Adv.* 10, 42157-42163 (2020). Some of them also reported the controlled-phase growth of SnSe₂. According to the data presented at present, the growth method is very similar to the previous reports.

Reply: As mentioned in these references by the reviewer, all of the works select similar precursors, *i.e.*, SnI₂, owing to its lower melting point to ensure the ease of synthesizing SnSe₂. However, we emphasized the essential role of the **substrate surface or termination**. The potassium ions enhance the charge transfer at the interface with 2D materials and induce interfacial interactions, which stabilize the high-order stacking polytypes. It is the key finding of our work. To prove our hypothesis, we examined the phases of SnSe₂ grown on a sapphire substrate under identical conditions. Notably, **the stacking polytypes of the SnSe₂ crystals grown on sapphire were entirely different from those observed on the mica substrate** (Supplementary Fig. 25).

Supplementary Fig. 25. The annular dark field scanning transmission electron microscopy (ADF-STEM) image of SnSe₂ (thickness = 474 nm) on a sapphire substrate. a, A low-

magnification ADF-STEM image showing atomic configuration of SnSe₂ slabs. b, SnSe₂ crystals with mixed AA (O), AB' (R'), and AC' (L') stacking sequences. c, Statistical analysis of the various stacking polytypes of SnSe₂ crystals grown on sapphire substrates. Scale bars: 100 nm in (a); 2 nm in (b).

2. The authors were only able to achieve partial AA and AB'-stacked SnSe₂ on one substrate, and this work would be of greater significance if complete AB'-stacked SnSe₂ growth could be attained.

Reply: In our experiments, we successfully achieved large-area SnSe₂ crystals with both AA and AB' stacking configurations. We conducted a Fourier transform (FFT) analysis on the obtained large-area STEM images and found **no super diffraction spots in the reciprocal space**, confirming the single-phase without stacking faults, which can also be clearly demonstrated by the atomic-resolution STEM images taken along the side view. The corresponding descriptions of the complete stacking orders have been provided, as shown hereafter.

In lines 190-193, relevant discussions were added. "To confirm the large-scale consistency of the stacking sequence, we performed low-magnification ADF-STEM imaging along with corresponding FFT pattern analysis. These results clearly demonstrate the successful synthesis of pure AA- and unconventional AB'-stacked SnSe₂ crystals (Supplementary Fig. 14)." Corresponding revisions have also been marked in red font in the revised manuscript.

Supplementary Fig. 14. The ADF-STEM images and the corresponding FFT patterns over a large area. a, The ADF-STEM image of AA-stacked SnSe₂ with a thickness of 27.8 nm (left panel) and the corresponding FFT pattern (right panel). The FFT results are in accord with the simulated SAED pattern. b, The ADF-STEM image of AB'-stacked SnSe₂ with a thickness of 26.1 nm and the relative FFT pattern. Analogously, the FFT results are also consistent with simulated SAED results. Scale bars: 5 nm (left panel) in (a, b); 2 nm⁻¹ (right panel).

3. Following the previous question, although the authors have presented several interesting phases through TEM data, there is a lack of discussion on the controllability and occurrence probability of these phases like 6R, 12R, 18R, 18C, etc., in SnSe₂. If these phases can only be observed in the sample without being reproducibly prepared over a large area, their

significance would be significantly diminished.

Reply: The 6R, 12R, 18R and 18C stacking polytypes, featuring high-order superlattices, are challenging to clearly distinguish by SHG or Raman spectroscopy. It can only be precisely confirmed by atomic resolution ADF-STEM, which is not a static characterization tool. Based on our statistical ADF-STEM results and DFT calculation results, gas flow is able to control metastable high-order SnSe₂ superlattices from the perspective of kinetic. To give a direct interpretation, we have listed the detailed growth conditions of various stacking polytypes in Supplementary Table 9.

Stacking sequence	SnI ₂ temperature	Gas flow (Ar)	Gas flow (H ₂)
AA stacked SnSe ₂	320 °C	20 sccm	0.4 sccm
AB' (2H) stacked SnSe ₂	340 °C	20 sccm	0.4 sccm
6R stacked SnSe ₂	340 °C	40 ~ 80 sccm	0.8 sccm
18R stacked SnSe ₂	340 °C	40 ~ 80 sccm	0.8 sccm
12R stacked SnSe ₂	340 °C	40 ~ 80 sccm	0.8 sccm
18C stacked SnSe ₂	340 °C	40 ~ 80 sccm	0.8 sccm

Supplementary Table 9. The detailed growth conditions of SnSe₂ crystals with different stacking sequence.

Additionally, in the method section, only changing gas flow can obtain different SnSe₂ stacking, the underlying mechanism remains incomprehensible.

Reply: Based on the DFT calculations, the symmetry operations R and L have identical formation energy. Consequently, the controlled growth of SnSe₂ crystals with various stacking sequences via purely thermodynamic control is challenging. However, upon examining the interactions with the substrate, it is noted that the interlayer energy difference associated with the AB' (R') and BA' (L') symmetry operations escalates to 0.601 eV (Fig 4k). This enhancement significantly increases the feasibility of experimentally manipulating the stacking configurations of the material. Consequently, the stacking structure of materials could be controlled through kinetic and thermodynamic processes experimentally. The corresponding results have been revised in Fig. 4. Supplementary Table 8. has been added to demonstrate the mechanism. Supplementary Note 4 is used to explain the detailed growth mechanism.

In lines 334-337, relevant discussions were added. “When the mica substrate is taken into account, electron transfer from the mica to the SnSe₂ crystals disrupts the energy degeneracy, resulting in an interlayer energy difference of 0.601 eV. These results lead to the stabilization of higher-order SnSe₂ stacking polytypes (Fig. 4k and Supplementary Table 8).” Corresponding revisions have also been marked in red font in the revised manuscript.

In lines 409-423, relevant discussions were presented. “**Supplementary Note 4: Strategy for the controlled synthesis of diverse SnSe₂ stacking registries with highly ordered superlattices.**”

During the synthesis of SnSe₂, we adjusted the Ar gas flow between 40 - 80 sccm to modulate the

nucleation and growth dynamics, aiming to reconfigure SnSe₂ slabs. In principle, the gas flow rate should be suitable to give viscous laminar flow to ensure a homogeneous growth environment during the reaction process. However, there is always a velocity gradient of gases in the growth chambers, and decreases to zero when near the substrate, presenting a stagnant layer⁴. Choosing the high-rate gas flow is more likely to generate turbulence, which is concluded from the definition of the Reynolds coefficient. Meanwhile, the temperature gradient inside the furnace also accelerates the formation of turbulence. During the process, the collision of the atoms will be enhanced, which breaks the thermodynamic equilibrium and transforms the system into kinetic dominance^{5,6}. The CVD growth process is generally complicated involving the sublimation and diffusion process of multiple precursors, maintaining the repeatability of 2D materials is still a big challenge⁷. Molecular beam epitaxy (MBE), owing to the high accuracy of controlling growth parameters, facilitates the precise manipulation of kinetic and thermodynamic behaviors to regulate the crystal symmetry in a variety of 2D materials⁸. ” Corresponding revisions have also been marked in red font in the revised supplementary Information.

Figure 4. A spectrum of stacking polytypes induced by a variety of interlayer symmetry operations in SnSe₂. (a) Top and side views of monolayer SnSe₂. The armchair [120] and zigzag [110] directions are marked with the pink arrow and dark blue arrow, respectively. (b) Atomic

models of bilayer SnSe₂ showing five distinct symmetry operations in sequence, *i.e.*, gliding $\frac{1}{\sqrt{3}} a$ from A to B defined as R, gliding $\frac{1}{\sqrt{3}} a$ from A to C defined as L, interlayer gliding from A to B combining with inversion named as R', interlayer gliding from A to C combining with inversion behaviors named as L'. Cross-sectional ADF-STEM images of SnSe₂ superlattices showing 6R ($R\bar{3}m$) (c), 18R ($R\bar{3}m$) (d), 18R (R3m) (e), and 18 (Cm) (f) space groups. The red and yellow false colors were used to label the gliding direction towards the right hand side direction [120] and left hand side [$\bar{1}20$], respectively. (g-j) Enlarged ADF-STEM images extracted from (c-f) presenting the unique stacking orders, respectively. Corresponding FFT patterns and simulated FFT patterns were depicted on the right panels. (k) **Calculated interlayer energies of R' and L' operation symmetry in b.** Theoretical calculated $\chi^{(2)}$ elements of 6R ($R\bar{3}m$) (l), 18R ($R\bar{3}m$) (m), 18R (R3m) (n) stacked SnSe₂. Scale bars: c-f, 1 nm; h-j, Left, 0.5 nm; Right, 2 nm⁻¹.

	Total energy (eV)	Interlayer energy (eV)	Formation energy (eV)
BA'-stacked SnSe ₂ /mica (L')	-3283.496	-38.486	-7.708
AB'-stacked SnSe ₂ /mica (R')	-3284.557	-39.087	-8.604

Supplementary Table 8. The interlayer energy as well as formation energy between R' and L' operations after taking into account of mica substrate.

4. The calibration of the current orientations of the SnSe₂ islands has some uncertainty. In Figure 2, the author employs red and yellow dotted lines to represent two orientations; however, it is evident that the edges of many triangles significantly deviate from the provided yellow dotted lines, indicating an estimated deflection of approximately 10 degrees.

Reply: We have identified a significant epitaxial relationship between the material and the substrate using a revised customized coding to eliminate the angle deflection. As illustrated in the Fig. 2a, we examined the orientation of approximately 150 crystal domains using SEM imaging. Among these, 96% of the domains exhibited an orientation of 0 degrees, while 4 % had an orientation deviating by 30 degrees. To demonstrate the epitaxial relationship, we statistically analyzed the growth orientations of crystal domains under varying growth conditions across different regions. It was observed that all crystal domains consistently exhibited two distinct orientational relationships. For example, the orientation analysis of the crystals grown on the mica substrate (H₂ = 0.5 sccm, Ar = 20 sccm) reveals that the domains with 0° and 30° alignments occur with probabilities of 100%, 98%, 99%, 100%, 99% and 100%, respectively (Supplementary Fig. 16). Besides, when modulating the growth parameters (H₂ = 1 sccm, Ar = 30 sccm), we have also achieved the perfect epitaxial growth. The aligned triangular flakes actually have two orientations with 30° (97%, 93% and 97%) rotation.

Consequently, SnSe₂ on mica has been successfully synthesized with two preferential orientations of either 0° or 60° (Supplementary Fig. 17).

In lines 199-203, relevant discussions were added. “After surveying a few hundred flakes, we found that (Fig. 2a) the as-grown triangular SnSe₂ domains predominantly manifest two distinct orientations (0° and 30°) with regards to the mica substrate, corresponding to 96% (0°) and 4% (30°) over the total population, respectively, validating the strong epitaxial relationship between the mica substrate and SnSe₂ crystals (Fig. 2b, c, Supplementary Fig. 16 and 17).” Corresponding revisions have also been marked in red font in the revised manuscript and revised supplementary Information.

Figure 2. Underlying mechanisms for growing SnSe₂ with novel phases. (a) Scanning Electron Microscopy (SEM) image showing epitaxial growth of SnSe₂ flakes on mica substrate. (b) The statistical counting showing the distribution of 0° and 30° twisted SnSe₂ flakes grown on mica. (c) Schematic illustration depicting SnSe₂ flakes grown on mica substrate with different crystallographic orientations. (d) The atomic resolution cross-sectional ADF-STEM image of AB'-stacked SnSe₂/mica substrate along the [1 $\bar{1}$ 0] zone axis. (e) The zoom-in ADF-STEM image reveals atomic distortion as depicted by the white dashed lines along the interface. (f) Corresponding FFT pattern from (e). Simulated FFT patterns of mica and AB'-stacking SnSe₂ were shown on the right panels. (g) The simulated STEM-ADF images derived from the DFT-optimized AA- and AB'-stacked SnSe₂ on the mica surface. The right panel showing the differential charge density of AA-, and AB'-stacked SnSe₂ on mica substrate. The area in orange is the region that gained electrons, and the area in green is the region that lost electrons. (h) DFT calculated formation energies of AA-stacked SnSe₂ and AB'-stacked SnSe₂ before and after considering the interface effect, respectively. Scale bars: a, 10 μ m; b, e, 1 nm; f, 2 nm⁻¹.

Supplementary Fig. 16. SEM images showing SnSe₂ crystals with well-defined epitaxial growth relationship (H₂ = 0.5 sccm, Ar = 20 sccm). a-c, The orientation of SnSe₂ flakes with 0° and 30°, respectively. d-f, The statistical results of flakes orientations. Scale bars: 10 μm in (a-c).

Supplementary Fig. 17. SEM image of SnSe₂ crystals grown at 600 °C (H₂ = 1 sccm, Ar = 30 sccm). Six regions are selected to count the amount of aligned SnSe₂ flakes. The alignment directions are labelled as 0° (pink line) and 30° (yellow line), respectively. Scale bars: 50 μm in (a-c, f), 100 μm in (d, e).

In general, I strongly concur with the author's perspective that symmetry regulation plays a crucial role in two-dimensional materials. However, the data provided by the authors in the current version does not substantiate their ability to effectively synthesize two-dimensional materials with controllable phases.

Reply: Thanks for your comments. We have added additional data and detailed growth conditions about the controllable synthesis of SnSe₂ as suggested by the referee with various stacking symmetries and emphasized the importance of substrate through epitaxial growth phenomenon.

Reviewer #2:

This manuscript presents a significant advancement in the field of 2D materials and crystal engineering. The authors have successfully demonstrated a novel method for controlling interlayer symmetry operations, which has potential implications for a wide range of applications. The study is well-executed, and the findings are supported by robust experimental and theoretical evidence. However, some sections could benefit from more concise writing to improve readability. Overall, the manuscript makes a valuable contribution to the field and is suitable for publication after minor revisions for clarity and conciseness.

Reply: We thank the reviewer for the careful review and constructive suggestions. As suggested by the referee, we have made relevant changes and added additional discussions regarding the controllability of stacking orders of SnSe₂ crystals and the significance of substrate as appended below. Additionally, **we have refined the language throughout the manuscript to enhance readability**. This involved replacing certain terms and eliminating redundant sentences. All corresponding revisions have been highlighted in red in the updated manuscript.

1. Is the interlayer sliding mentioned in the article controllable?

Reply: As for the minor interlayer gliding behaviors in AB'-stacked SnSe₂, DFT calculations exhibit that the energy difference of AB' stacking sequence with 0.22 Å is just 1.543 meV/f.u., **approaching the uncertainty of DFT calculations**. Therefore, it is challenging to achieve the modulation of AB'-stacked SnSe₂ crystals with minor interlayer gliding behaviors experimentally. The corresponding DFT calculation results are shown in Supplementary Table 1.

Stacking orders	E (eV)*	E (eV/f.u.)	ΔE (meV/f.u.)	Formation energy (eV/f.u.)
AB'/BC'/CA'	-24.773	-12.386	1	-0.121
AC'/BA'/CB'	-24.773	-12.386	1	-0.121
AA/BB/CC	-24.774	-12.387	0	-0.122
AB/BC/CA	-24.746	-12.373	14	-0.108
AC/BA/CB	-24.689	-12.344	43	-0.079
AA'/BB'/CC'	-24.749	-12.375	11.5	-0.110

* Total energy E (eV); f. u.= formula unit

Supplementary Table 1. The DFT calculated formation energies of SnSe₂ as a function of stacking order without accounting for the substrate effect.

Are the different high-order symmetries randomly generated?

Reply: Theoretically, the formation energy of symmetric operations R' (AB') and L' (BA') is in a degenerate state (Supplementary Fig. 35). However, when accounting for the charge transfer process between the mica substrate and the SnSe₂ material, the interlayer energy of two

operations showed -38.486 and -39.087 eV, respectively. The interlayer energy difference is 0.601 eV (Supplementary Table 8.), which is smaller than the energy difference observed between the AA and AB' stacking configurations (0.841 eV). Consequently, the thermodynamic window for experimentally controlling the stacking structure is quite limited.

The gas flow influences the nucleation sites during the growth process. High gas flow creates the turbulent environment, which benefits the stable synthesis of SnSe₂ crystals with high-order symmetries. Thus, we have summarized the detailed growth parameters provided in Supplementary Table 9.

In lines 332-334, relevant discussions were added. “For a comprehensive understanding of the growth dynamics, refer to Supplementary Note 4 and the detailed growth conditions based on modulating the gas flow in a range of 40 ~ 80 sccm to grow various stacking orders are listed in Supplementary Table 9.” Corresponding revisions have also been marked in red font in the revised manuscript and revised supplementary Information.

Supplementary Fig. 35. The calculated formation energy of the five symmetric operations without the consideration of mica substrate.

	Total energy (eV)	Interlayer energy (eV)	Formation energy (eV)
BA'-stacked SnSe ₂ /mica (L')	-3283.496	-38.486	-7.708
AB'-stacked SnSe ₂ /mica (R')	-3284.557	-39.087	-8.604

Supplementary Table 8. The interlayer energy as well as formation energy between R' and L' operations after taking into account of mica substrate.

Stacking sequence	SnI ₂ temperature	Gas flow (Ar)	Gas flow (H ₂)
AA stacked SnSe ₂	320 °C	20 sccm	0.4 sccm
AB' (2H) stacked SnSe ₂	340 °C	20 sccm	0.4 sccm
6R stacked SnSe ₂	340 °C	40 ~ 80 sccm	0.8 sccm
18R stacked SnSe ₂	340 °C	40 ~ 80 sccm	0.8 sccm
12R stacked SnSe ₂	340 °C	40 ~ 80 sccm	0.8 sccm
18C stacked SnSe ₂	340 °C	40 ~ 80 sccm	0.8 sccm

Supplementary Table 9. The detailed growth conditions of SnSe₂ crystals with different stacking sequence.

2. What were the reference materials or benchmarks used to validate the SHG measurements?

Reply: We performed SHG measurements on bilayer graphene and monolayer MoS₂ as reference materials, with the former exhibiting no SHG signal and the latter displaying a clear SHG response. These references were used to validate the SHG signals observed in our samples. As illustrated in Supplementary Fig. R1, Raman spectra of graphene with the intensity of 2D and G peak is approximately 1, demonstrating the bilayer feature of as-exfoliated graphene. Meanwhile, the Raman shift difference between E_{2g} and A_{1g} peaks was 20 cm⁻¹, a fingerprint feature demonstrating the monolayer nature of the MoS₂ flake (*Nat. Commun.* **2020**, *11*, 1011.). Besides, the SHG intensity polar plot with six-fold SHG intensity pattern.

Supplementary Fig. R1. The comparison of SHG intensity of monolayer MoS₂, bilayer graphene and as-grown SnSe₂ crystals. a, The Raman spectra of bilayer graphene exfoliated on SiO₂/Si substrate. Illustration showing the corresponding optical image. b, The Raman spectra of

monolayer MoS₂ grown on SiO₂/Si substrate. Illustration showing the CVD-grown monolayer MoS₂ flake. c, The polar plot of monolayer MoS₂, demonstrating the six-fold symmetry. d, SHG spectra of MoS₂, graphene and SnSe₂ flake. Scale bar: 4 μm.

3. Why mica substrate was used in the study and other substrates (e.g. sapphire, MgO) are not available?

Reply: As depicted in Supplementary Fig. 20, the surface of mica is predominantly composed of potassium ions. Consistent with previous reports, the surface of all mica layers exhibits a positive charge (*Sci. Rep.* **2023**, *13*, 7880.). This character facilitates charge transfer between crystals and mica substrate. Detailed in Fig. 2d, differential charge density calculations reveal significant charge transfer between the surface. Specifically, 10.71 e⁻ are transferred from the potassium atoms on the mica to bridging Se atoms in AB'-stacked SnSe₂, while a slightly lower transfer of 10.44 e⁻ in AA-stacked SnSe₂ (Fig. 2g). Initially, the formation energy between AA and AB'-stacked SnSe₂ is minimal, at only 1 meV, indicating a challenge in controlling the stacking arrangement of SnSe₂ crystals solely by adjusting the synthesis temperature of SnI₂. However, considering the charge transfer dynamics between the mica substrate and the layered materials, the formation energy differential between the AB' and AA stacking configurations increases to 115 meV (Fig. 2h). This adjustment significantly enhances the ability to selectively synthesize SnSe₂ with different structural arrangements, highlighting the pivotal role of substrate-induced charge interactions in dictating the stacking order and crystalline structure of layered materials.

The typical c-plane oriented sapphire generally exhibits an aluminium (Al) surface termination, as depicted in Supplementary Fig. 23. Experimentally, we observed that SnSe₂ crystals did not exhibit perfect epitaxial alignment with the sapphire substrate under the same growth conditions (Supplementary Fig. 24, 25). Specifically, ADF-STEM imaging revealed a significant off-axis deviation of the sapphire substrate when SnSe₂ was aligned along its principal axis (Supplementary Fig. 26). For SnSe₂ crystals with a thickness of 474 nm, we qualified the stacking behaviors, finding that 75% exhibited BA' (L') stacking behaviors, 3.6 % with AB' (R') stacking orders and 21.4 % had intrinsic stacking slabs (Labelled as O).

Bader charge analysis further revealed that the charge transfer between the sapphire substrate and SnSe₂ was significantly weakened compared to that observed with mica. This diminished interfacial interaction highlights the substrate's role in modulating SnSe₂ stacking behaviors, as the stronger charge transfer with mica corresponds to different stacking and structural characteristics. Furthermore, the work function of sapphire is approximately 6.0 eV, closely matching that of SnSe₂, whereas the variation in formation energy across different substrates can reach up to 4.0 eV, underscoring the critical influence of substrate choice in the material growth process (Supplementary Table 2 and 4).

Thus, we have added additional DFT calculations based on AA-stacked and AB'-stacked SnSe₂ on sapphire substrate and mica substrate, respectively. The corresponding descriptions have been provided, as shown hereafter.

In lines 231-236, relevant discussions were added. “On the contrary, pure AB' stacking was not observed on sapphire substrates (Supplementary Fig. 22-26). Upon characterizing the stacking behaviors, we found that 75% exhibited BA' stacking behaviors, 3.6 % with AB' stacking orders and 21.4 % had intrinsic stacking slabs. The weakened charge transfer process between sapphire and SnSe₂ crystals results in only a 28 meV difference in formation energy difference, highlighting the critical role of the substrate (Supplementary Fig. 27, Supplementary Table 4, 5 and Supplementary Note 2).” Corresponding revisions have also been marked in red font in the revised manuscript.

In lines 334-337, relevant discussions were added. “When the mica substrate is taken into account, electron transfer from the mica to the SnSe₂ crystals disrupts the energy degeneracy, resulting in an interlayer energy difference of 0.601 eV. These results lead to the stabilization of higher-order SnSe₂ stacking polytypes (Fig. 4k and Supplementary Table 8).” Corresponding revisions have also been marked in red font in the revised manuscript.

In lines 378-391 in revised supplementary Information, relevant discussions were added. **“Supplementary Note 2: The calculation results of SnSe₂ crystals on sapphire substrate.**

The typical c-plane oriented sapphire generally exhibits an aluminum (Al) surface termination, as depicted in Supplementary Fig. 23. Experimentally, SnSe₂ crystals did not exhibit perfect epitaxial alignment with the sapphire substrate under the same growth conditions (Supplementary Fig. 22). Specifically, ADF-STEM imaging revealed a significant off-axis deviation of sapphire substrate when SnSe₂ was aligned along its principal axis (Supplementary Fig. 26). For SnSe₂ crystals with a thickness of 474 nm, we qualified the stacking behaviors, finding that 75% exhibited BA' (L') stacking behaviors, 3.6 % with AB' (R') stacking orders and 21.4 % had intrinsic stacking slabs (Labelled as O).

Bader charge analysis further revealed that the charge transfer between the sapphire substrate and SnSe₂ was significantly weakened compared to that observed with mica (2.03 e⁻ for AA stacked SnSe₂ and 2.02 e⁻ for AB'-stacked SnSe₂). Furthermore, the work function of sapphire is approximately 6.0 eV, closely matching that of SnSe₂, whereas the variation in formation energy across different substrates can reach up to 4.0 eV, underscoring the critical influence of substrate choice in the material growth process (Supplementary Table 4).” Corresponding revisions have also been marked in red font in the revised supplementary Information.

Supplementary Fig. 20. The atomic structure of mica substrate. a, b, The ADF-STEM image of mica substrate. Each sheet consists of two tetrahedral silicate layers sandwiching an octahedral sheet containing aluminium or magnesium with the layers held together by positively charged potassium. c, The simulated ADF-STEM image based on the mica atomic model. Scale bars: 1 nm in (a), 0.5 nm in (b).

Supplementary Fig. 22. SEM images of SnSe₂ crystal on sapphire substrate. The crystals grown on the sapphire substrate were free of a significant epitaxial relationship with the underlying substrate. Scale bars: 200 μm in (a, b); 20 μm in (c).

Supplementary Fig. 23. The atomic structure of sapphire with lattice constant $a = b = 4.8050$

\tilde{a} , $c = 13.1163 \text{ \AA}$, $\alpha = \beta = 90^\circ$, $\gamma = 120^\circ$. The red and blue ball corresponds to Al and O atoms, respectively. The Al cations are arranged in a tightly packed lattice with O anions occupying the spaces between them, creating a repeating pattern of alternating layers of Al and O atoms throughout the crystal.

Supplementary Fig. 25. The ADF-STEM image of SnSe₂ (thickness = 474 nm) on a sapphire substrate. a, A low-magnification ADF-STEM image showing atomic configuration of SnSe₂ slabs. b, SnSe₂ crystals with mixed AA (O), AB' (R'), and AC' (L') stacking sequences. c, Statistical analysis of the various stacking polytypes of SnSe₂ crystals grown on sapphire substrates. Scale bars: 100 nm in (a); 2 nm in (b).

Supplementary Fig. 26. The atomic structure at the interface between SnSe₂ and sapphire substrate. a, ADF-STEM image showing the atomic arrangement along the sapphire [010] zone axis. b, c, Zoom-in ADF-STEM images providing SnSe₂ and sapphire substrate from a, respectively. d, The arrangement of SnSe₂ crystals along the $[1\bar{1}0]$ Zone axis. Panels b and d indicate a lack of lattice matching between SnSe₂ and sapphire substrate. e, The ADF-STEM image along the $[\bar{1}10]$ zone axis of sapphire. f, g, Zoom-in ADF-STEM images of SnSe₂ and Al₂O₃ crystals. h, Atomic structure of SnSe₂ along [100] zone axis. Scale bars: 5 nm in (a, e), 1 nm in (b, c), 10 nm in (d, h), 2 nm in (f, g).

Supplementary Fig. 27. DFT calculations of the interlayer energy and the formation energy for different stacking structures. a, b, The simulated ADF-STEM images derived from the DFT-optimized AA- and AB'-stacked SnSe₂ on the sapphire surface. The right panel illustrates the differential charge density of both AA-, and AB'-stacked SnSe₂ on the sapphire substrate. The area in green is the region that gained electrons, and the area in purple represent that lost electrons. c, The interlayer energy difference and the formation energy difference between AA and AB'-stacked SnSe₂ on the sapphire substrate. d, The interlayer energy of AA and AB'-stacked SnSe₂ on sapphire and mica substrate, respectively.

Work function (eV)	AA stacking orders	AB' stacking orders
Mica	3.049	3.0512
SnSe ₂	5.888	5.9149

Supplementary Table 2. DFT results showing work functions of mica crystal, AA- and AB'-stacked SnSe₂ crystals with the mica/SnSe₂ system.

	Total energy (eV)	Substrate energy (eV)	SnSe ₂ energy (eV)	Interlayer energy (eV)
AA-stacked SnSe ₂ /Sapphire	-2212.726	-1903.664	-296.501	-12.561
AB'-stacked SnSe ₂ /Sapphire	-2213.862	-1904.770	-296.521	-12.571
AA-stacked SnSe ₂ /Mica	-3283.314	-2457.692	-787.375	-38.246
AB'-stacked SnSe ₂ /Mica	-3284.557	-2458.615	-786.855	-39.087

Supplementary Table 3. The interlayer energy of AA and AB'-stacked SnSe₂ on sapphire and

mica substrate.

Work function (eV)	AA stacking orders	AB' stacking orders
Sapphire	6.040	6.154
SnSe ₂	5.606	5.921

Supplementary Table 4. Work functions of sapphire, AA stacked and AB'-stacked SnSe₂ crystals in sapphire/SnSe₂ system.

	Formation energy (eV)
AA-stacked SnSe ₂ /Sapphire	-3.208
AB'-stacked SnSe ₂ /Sapphire	-3.236
AA-stacked SnSe ₂ /Mica	-7.949
AB'-stacked SnSe ₂ /Mica	-8.064

Supplementary Table 5. The formation energy of AA and AB'-stacked SnSe₂ on sapphire and mica substrate.

4. Although the research results indicate that different stacking sequences and symmetries affect the second harmonic generation (SHG) signal, most of the findings rely on theoretical calculations and simulations.

Reply: Experimentally, the difference in symmetry caused by small interlayer slips and interlayer flip is difficult to be detected by SHG and Raman spectroscopy. Consequently, we performed lots of SHG calculations for SnSe₂ with various stacking polytypes. The calculated results indicated that the SHG response of SnSe₂ crystals with 6R and 18R symmetries are remarkably similar.

Experimental validation is primarily focused on a limited number of samples, lacking experimental data support under broader conditions and materials.

Reply: To comprehensively demonstrate the SHG signals across various samples, we systematically investigated the SHG response of SnSe₂ domains with non-centrosymmetric structures. To experimentally validate the structural universality, SHG measurements were performed on 15 distinct domains, and their internal structural homogeneity was assessed through SHG mapping. Additionally, the samples were transferred from mica substrates to SiO₂/Si substrate. SHG measurements over a large area were conducted on SiO₂/Si and mica substrates, respectively. As shown in Supplementary Fig. 9 and 10, the domains across this extensive area exhibit a significant SHG response, with a 100% detection rate. The corresponding experiment data have been added in the revised supplementary Information.

Supplementary Fig. 9. The SHG signals of SnSe₂ crystals with broken centrosymmetry on a mica substrate. a, An optical image of 15 SnSe₂ flakes dispersed across the mica substrate. b, SHG signals obtained from all flakes, with each exhibiting a 100% SHG response. Scale bar: 20 μm in (a).

Supplementary Fig. 10. SHG signals of SnSe₂ domains transferred onto Au film on Si substrate. a, b SHG mapping over a large area with 100% SHG signal coverage. c, SHG results from 15 individual flakes, demonstrating the controllability of AB¹-stacked SnSe₂ crystals.

5. In addition to the optical anisotropy characterizations mentioned in the article, other optical anisotropy characterizations (e.g., Raman spectroscopy) can be added to the Supplementary Note.

Reply: In addition to SHG anisotropic characterization, we have performed polarized Raman spectroscopy on AB¹-stacked SnSe₂ crystals. As shown in Supplementary Fig. 38b and c, the intensities of the E_{2g} and A_{1g} modes remain overall constant. To further validate the reliability of the experimental results, we performed a polarized Raman spectrum on NbOCl₂ crystal as a reference. Notably, the intensity of the 667 cm⁻¹ peak exhibits a clear angular dependence, decreasing from a maximum at 0° to zero at 90°, and then gradually increasing from 90° to 180°. The corresponding experiment data have been added in revised manuscript and supplementary Information.

In lines 350-352, relevant discussions were added. “In addition, it is noteworthy that additional optical anisotropic characterization such as polarized Raman spectroscopy on AB¹-stacked SnSe₂ crystals, and the results are presented in Supplementary Note 6.” Corresponding revisions have also been marked in red font in the revised manuscript.

In lines 451-456, relevant discussions were added in supplementary Information. **“Supplementary Note 6: Polarization dependence of Raman spectra of the AB'-stacked SnSe₂**

As shown in Supplementary Fig. 38b and c, the intensities of E_{2g} and A_{1g} mode are basically as circles no matter what directions were placed. To further validate the reliability of the experimental results, we performed a polarized Raman spectrum on NbOCl₂ crystal as a reference. Notably, the intensity of the 667 cm⁻¹ peak exhibits a clear angular dependence, decreasing from a maximum at 0° to zero at 90°, and then gradually increasing from 90° to 180°.”

Supplementary Fig. 38. Polarization dependence of Raman spectra of the AB'-stacked SnSe₂ with the 532 nm excitation in parallel polarization configuration. a, Schematic illustrations of polarized Raman measurements. b, Angle-dependent polarized Raman intensity of AB'-stacked SnSe₂ crystal. Polar plot of Raman intensity for typical E_{2g} mode (c) and A_{1g} mode (d). e, Raman spectra at various polarization angles under parallel configuration. f, Polarization-dependent Raman intensity of 667 cm⁻¹ in a NbOCl₂ crystal.

Reviewer #3:

In this work, Han *et al.* studied SnSe₂ grown on mica by a CVD method that utilizes SnI₂ and Se as the precursors. They proposed a growth mechanism to explain how potassium terminated on mica could stabilize the metastable AB'-stacked SnSe₂ through the interlayer charge transfer. In the second part of the manuscript, the authors focused on the atomic analysis of ADF-STEM images of AB'-SnSe₂ multilayers, including some of the stacking polytypes and their NLO properties. While a lot of interesting data were presented and the results are relatively new compared to similar SnSe₂ work, I don't fully agree with their conclusions. The mica they used indeed stabilized some of the SnSe₂ domains into AB'-stacked layers and helped epitaxy. However, I don't think it can guide SnSe₂ to grow into a particular stacking polytype.

Reply: Firstly, the surface of the mica substrate is essential in the growth of SnSe₂ crystals with various stacking polytypes. The similarity in work functions between the mica substrate and SnSe₂ crystals facilitates the charge transfer between them. Specifically, the mica substrate donates 10.44 e⁻ for AA-stacked SnSe₂ and 10.71 e⁻ for AB'-stacked SnSe₂. The variation in the number of transferred electrons results in an increase in formation energy difference between the AA and AB' stacking configurations, rising from 1 meV to 115 meV. Consequently, the thermodynamic control over the growth temperature can be effectively employed to modulate the formation of these two stacking structures.

For comparison, we also employed the sapphire substrate to grow SnSe₂ and found that the stacking behaviors of the sapphire substrate preferred to be mixed stacking orders rather than pure AA and AB' stacking sequence. Meanwhile, DFT results indicate that the formation energy difference was only 28 meV, which poses challenges for the precise modulation of stacking behaviors using thermodynamic methods. The corresponding experiment data and description have been added in the revised manuscript and supplementary Information.

In lines 222-225, relevant discussions were added. "Differential charge density calculations suggest that there is substantial charge transfer along the interface, with 10.71 e⁻ being transferred from the K atoms to bridging Se atoms in AB'-stacked SnSe₂, and 10.44 e⁻ in AA-stacked SnSe₂ (Fig. 2g and Supplementary Table 2)." Corresponding revisions have also been marked in red font in the revised manuscript.

In lines 231-236, relevant discussions were added. "On the contrary, pure AB' stacking was not observed on sapphire substrates (Supplementary Fig. 22-26). Upon characterizing the stacking behaviors, we found that 75% exhibited BA' stacking behaviors, 3.6 % with AB' stacking orders and 21.4 % had intrinsic stacking slabs. The weakened charge transfer process between sapphire and SnSe₂ crystals results in only a 28 meV difference in formation energy difference, highlighting the critical role of the substrate (Supplementary Fig. 27, Supplementary Table 4, 5 and Supplementary Note 2)." Corresponding revisions have also been marked in red font in the revised manuscript.

Supplementary Fig. 27. DFT calculations of the interlayer energy and the formation energy for different stacking structures. a, b, The simulated ADF-STEM images derived from the DFT-optimized AA- and AB'-stacked SnSe₂ on the sapphire surface. The right panel illustrates the differential charge density of both AA-, and AB'-stacked SnSe₂ on the sapphire substrate. The area in green is the region that gained electrons, and the area in purple represent those lost electrons. c, The interlayer energy difference and the formation energy difference between AA- and AB'-stacked SnSe₂ on the sapphire substrate. d, The interlayer energy of AA- and AB'-stacked SnSe₂ on sapphire and mica substrate, respectively.

	Total energy (eV)	Substrate energy (eV)	SnSe ₂ energy (eV)	Interlayer energy (eV)
AA-stacked SnSe ₂ /Sapphire	-2212.726	-1903.664	-296.501	-12.561
AB'-stacked SnSe ₂ /Sapphire	-2213.862	-1904.770	-296.521	-12.571
AA-stacked SnSe ₂ /Mica	-3283.314	-2457.692	-787.375	-38.246
AB'-stacked SnSe ₂ /Mica	-3284.557	-2458.615	-786.855	-39.087

Supplementary Table 3. The interlayer energy of AA and AB'-stacked SnSe₂ on sapphire and mica substrate.

	Formation energy (eV)
AA-stacked SnSe ₂ /Sapphire	-3.208
AB'-stacked SnSe ₂ /Sapphire	-3.236

AA-stacked SnSe ₂ /Mica	-7.949
AB'-stacked SnSe ₂ /Mica	-8.064

Supplementary Table 5. The formation energy of AA and AB'-stacked SnSe₂ on sapphire and mica substrate.

Fig. 2 shows the distribution of 0- and 30-degree epi-SnSe₂ on mica. However, the authors did not correlate how many of the 0 degrees and how many of the 30 degrees are AB' stacking. I think that AB' type flakes must have a unique epitaxial relationship with mica to have the interlayer effect discussed in Fig. 2g. (Perhaps the authors can come out with a rapid SHG characterization method to examine a number of domains orienting at both 0- and 30-degree on mica.)

Reply: The optical images presented in Fig. 2 are intended to illustrate the epitaxial relationship between the material and the substrate. Both AA and AB'-stacked SnSe₂ crystals exhibit obvious interactions with the substrate, as evidenced by experimental observations (Supplementary Fig. 16 and 17). Thus, the stacking sequence of the SnSe₂ crystal was not demonstrated through the epitaxial relationship with the substrate.

However, based on the optical images obtained from numerous SnSe₂ flakes, we found that certain AB'-stacked SnSe₂ tend to adopt a rounded shape, which can be attributed to dynamic growth conditions. This characteristic can be considered as a distinguishing feature for distinguishing different stacking polytypes. From the perspective of thermodynamic and kinetic growth principles, stacking sequences other than the AA stacking order preferred to be round shapes (Supplementary Fig. 2 and 8).

In lines 125-127, relevant discussions were added. "On the other hand, when the growth temperature was elevated to 340 °C reaching a metal-rich condition, the edges become rounded (Fig. 1b and Supplementary Fig. 2), suggesting a nonequilibrium growth condition between dynamics and thermodynamics⁴⁵." Corresponding revisions have also been marked in red font in the revised manuscript.

Supplementary Fig. 2. The optical image of AB'-stacked SnSe₂ crystals with irregular shapes marked by pink and yellow dashed lines.

Supplementary Fig. 8. The SHG characterizations of SnSe₂ domains with obvious SHG response. a, The optical image of rounded triangle and rounded SnSe₂ domains. b, The SHG intensity of SnSe₂ flakes indicated in panel a. Insert image showing the corresponding SHG intensity.

Supplementary Fig. 10. SHG signals of SnSe₂ domains transferred onto Au film on Si substrate. a, b SHG mapping over a large area with 100% SHG signal coverage. c, SHG results from 15 individual flakes, demonstrating the controllability of AB'-stacked SnSe₂ crystals.

Supplementary Fig. 16. SEM images showing SnSe₂ crystals with well-defined epitaxial growth relationship (H₂ = 0.5 sccm, Ar = 20 sccm). a-c, The orientation of SnSe₂ flakes with 0° and 30°, respectively. d-f, The statistical results of flakes orientations. Scale bars: 10 μm in (a-c).

Supplementary Fig. 17. SEM image of SnSe₂ crystals grown at 600 °C (H₂ = 1 sccm, Ar = 30 sccm). Six regions are selected to count the amount of aligned SnSe₂ flakes. The alignment directions are labelled as 0° (pink line) and 30° (yellow line), respectively. Scale bars: 50 μm in (a-c, f), 100 μm in (d, e).

Moving forward to Fig. 4, the authors found several stacking polytypes in ADF-STEM experiments and used interlayer sliding, gliding, and inversion to explain their formation. While these stacking polytypes are certainly interesting and provide great insights into the community, the authors cannot link their formation to the substrate-guided growth they proposed.

Reply: The corresponding interlayer energy calculations are provided in Fig. 4k. The interlayer energy of AB'-stacked SnSe₂ and BA'-stacked SnSe₂ was separated into about -8.064 eV and -7.708 eV, respectively. Consequently, we could modulate the stacking polytypes of SnSe₂ from a prospective of dynamics. Experimentally, choosing the high-rate gas flow is more likely to generate turbulence, which is concluded from the definition of the Reynolds coefficient.

Meanwhile, the temperature gradient inside the furnace also accelerates the formation of turbulence. During the process, the collision of the atoms will be enhanced, which breaks the thermodynamic equilibrium and transforms the system into kinetic dominance.

In lines 323-329, relevant discussions were added. “To unveil the underlying growth mechanism of different degrees of SnSe₂ superlattices, we quantified the interlayer energies of various symmetry operations. It can be seen that the interlayer energies of R' and L' operations are observed to be the most thermodynamically favored stacking orders as compared to R and L operations by 14 and 43 meV, respectively (Supplementary Fig. 35). When the mica substrate is taken into account, electron transfer from the mica to the SnSe₂ crystals disrupts the energy degeneracy, resulting in an interlayer energy difference of 0.601 eV. These results lead to the stabilization of higher-order SnSe₂ stacking polytypes (Fig. 4k and Supplementary Table 8).” Corresponding revisions have also been marked in red font in the revised manuscript.

In lines 332-334, relevant discussions were added. “For a comprehensive understanding of the growth dynamics, refer to Supplementary Note 4 and the detailed growth conditions based on modulating the gas flow in a range of 40 ~ 80 sccm to grow various stacking orders are listed in Supplementary Table 9.” Corresponding revisions have also been marked in red font in the revised manuscript.

Supplementary Fig. 35. The calculated formation energy of the five symmetric operations without the consideration of mica substrate. The energy of AB' and BA' stacking sequences are thermodynamically equivalent and represent the most stable stacking polytypes.

	Total energy (eV)	Interlayer energy (eV)	Formation energy (eV)
BA'-stacked	-3283.496	-38.486	-7.708

SnSe ₂ /mica (L')			
AB'-stacked SnSe ₂ /mica (R')	-3284.557	-39.087	-8.604

Supplementary Table 8. The interlayer energy as well as formation energy between R' and L' operations after taking into account of mica substrate.

Stacking sequence	SnI ₂ temperature	Gas flow (Ar)	Gas flow (H ₂)
AA stacked SnSe ₂	320 °C	20 sccm	0.4 sccm
AB' (2H) stacked SnSe ₂	340 °C	20 sccm	0.4 sccm
6R stacked SnSe ₂	340 °C	40 ~ 80 sccm	0.8 sccm
18R stacked SnSe ₂	340 °C	40 ~ 80 sccm	0.8 sccm
12R stacked SnSe ₂	340 °C	40 ~ 80 sccm	0.8 sccm
18C stacked SnSe ₂	340 °C	40 ~ 80 sccm	0.8 sccm

Supplementary Table 9. The detailed growth conditions of SnSe₂ crystals with different stacking sequence.

I also think it would be great if the authors linked these polytypes to any ferroelectric properties or showed some examples of property modulation.

Reply: To demonstrate the polar domain, piezoresponse force microscopy (PFM) measurements were conducted on AB'-stacked SnSe₂ crystals transferred on a conducted substrate (Au film on SiO₂/Si substrate). PFM displays that the presence of out-of-plane ferroelectricity in AB'-stacked SnSe₂ crystals. The corresponding experiment data and description have been added in revised supplementary Information.

In lines 193-196, relevant discussions were added. "Based on the demonstration of AB'-stacked SnSe₂, we conducted piezoresponse force microscopy (PFM). As illustrated in Supplementary Fig. 15, the observed phase hysteresis and butterfly shaped amplitude loops are an indication of polarization switching, demonstrating the presence of out-of-plane ferroelectricity in AB'-stacked SnSe₂ crystals." Corresponding revisions have also been marked in red font in the revised manuscript.

Supplementary Fig. 15. Piezoresponse force microscopy (PFM) characterizations of AB'-stacked SnSe₂ crystals. The AFM image (a), amplitude image and phase image of the corresponding domain (labelled as yellow dashed line). d, The domains in a with a significant SHG response. Relative PFM phase hysteresis (e) and butterfly loop (f) of nanosheet with ferroelectric properties. Scale bars: 2 μm in (a-c).

To make their formation controllable, layer number control capabilities and chemical potential balance between Sn and Se need to be established, such as by MOCVD [e.g., *Adv. Mater.* **2024**, *36*, 2400800, in which SnSe₂ that grows at 200 $^{\circ}\text{C}$ was reported].

Reply: According to the reference, the phase selective synthesis of SnSe₂ and SnSe have been achieved through controlling the growth temperature and mass flow ratio. As a result, we also add extra experiment to exhibit how to control the precursor mass and gas flow. Experimentally, by precisely controlling the temperature conditions, we were able to selectively synthesize high-quality SnSe and SnSe₂ crystals, respectively. The corresponding optical image and Raman spectra have been added in Supplementary Fig. 3. The synthesis of SnSe₂ and SnSe crystals with the modulation of the temperature of SnI₂. With the raising of growth temperature (750 $^{\circ}\text{C}$), SnSe₂ crystal transfer into SnSe flakes.

In terms of layer number control, we were able to vary the thickness of SnSe₂ crystals from 1.4 nm to several hundred nanometers. Supplementary Fig. 4. illustrates the correlation between layer numbers and the gas flow of H₂.

In lines 127-129, relevant discussions were added. “When the substrate temperature was further increased to 750 $^{\circ}\text{C}$, the obtained crystals predominantly consisted of SnSe flakes (Supplementary Fig. 3).” Corresponding revisions have also been marked in red font in the revised manuscript.

In lines 129-131, relevant discussions were added. “The thickness of the as-grown SnSe₂ flakes typically ranges from bilayer (~1.4 nm) to tens of nanometers through modulating the gas flow of H₂, as confirmed by atomic force microscopy (AFM) (Supplementary Fig. 4).” Corresponding revisions have also been marked in red font in the revised manuscript.

Supplementary Fig. 3. The Raman spectra of SnSe₂ (a) and SnSe (c) and their atomic structure model, respectively. Scale bar: 4 μm. As for SnSe₂ crystals, 117 cm⁻¹ and 184 cm⁻¹ corresponds to the E_g and A_{1g} vibration modes, respectively. As for SnSe, 71 cm⁻¹, 106 cm⁻¹, 127 cm⁻¹ and 147 cm⁻¹ correspond to the A_g¹, B_{3g}¹, A_g² and A_g³ vibration modes, respectively.

Supplementary Fig. 4. AFM topological image of as-synthesized SnSe₂ with H₂ gas flow at 0.4 sccm, 0.7 sccm and 1.5 sccm. a, c, e, AFM images of SnSe₂ crystals with different thickness. b, d, f, Optical images of SnSe₂ using different gas flow. Scale bars: 4 μm in (a, c, e), 50 μm in (b, f), 10 μm in (d).

Based on my observation, I feel that this work might be more suitable for other 2D material journals or should be returned to the authors for a major revision.

Reply: Thanks for your sincere suggestions. We have added additional experiments and theoretical calculations based on the controlled growth of SnSe₂ crystals with various stacking polytypes. DFT results emphasized the critical role of mica substrate, guiding the charge transfer process between SnSe₂ crystals. Thus, we hoped that the revised manuscript could be published on *Nature Communications*.